# Deletion upstream of *MAB21L2* highlights the importance of evolutionarily conserved non-coding sequences for eye development

Fabiola Ceroni [1,2,17], Munevver B. Cicekdal [3,4,5,17], Richard Holt[1,17], Elena Sorokina[6,17], Nicolas Chassaing [7,8], Samuel Clokie[1], Thomas Naert [3,9,10], Lidiya V. Talbot [1], Sanaa Muheisen[6], Dorine A. Bax[1], Yesim Kesim[1,11,12], Emma C. Kivuva [13], Catherine Vincent-Delorme[14], Soeren S. Lienkamp [9,10], Julie Plaisancié [7,8,15], Elfride De Baere [4,5], Patrick Calvas[7,8], Kris Vleminckx [3] ✉, Elena V. Semina [6] ✉ & Nicola K. Ragge [1,16] ✉

Anophthalmia, microphthalmia and coloboma (AMC) comprise a spectrum of developmental eye disorders, accounting for approximately 20% of childhood visual impairment. While non-coding regulatory sequences are increasingly recognised as contributing to disease burden, characterising their impact on gene function and phenotype remains challenging. Furthermore, little is known of the nature and extent of their contribution to AMC phenotypes. We report two families with variants in or near *MAB21L2*, a gene where genetic variants are known to cause AMC in humans and animal models. The first proband, presenting with microphthalmia and coloboma, has a likely pathogenic missense variant (c.338 G > C; p.[Trp113Ser]), segregating within the family. The second individual, presenting with microphthalmia, carries an ~ 113.5 kb homozygous deletion 19.38 kb upstream of *MAB21L2*. Modelling of the deletion results in transient small lens and coloboma as well as midbrain anomalies in zebrafish, and microphthalmia and coloboma in *Xenopus tropicalis*. Using conservation analysis, we identify 15 non-coding conserved elements (CEs) within the deleted region, while ChIP-seq data from mouse embryonic stem cells demonstrates that two of these (CE13 and 14) bind Otx2, a protein with an established role in eye development. Targeted disruption of CE14 in *Xenopus tropicalis* recapitulates an ocular coloboma phenotype, supporting its role in eye development. Together, our data provides insights into regulatory mechanisms underlying eye development and highlights the importance of non-coding sequences as a source of genetic diagnoses in AMC.

Anophthalmia (absent eye), microphthalmia (small eye) and coloboma (disruption of optic fissure closure) (AMC) form a phenotypic spectrum of developmental disorders affecting ~ 6–30 per 100,000 live births[1,2], and account for up to 20% of childhood visual impairment[2]. While genetic variants affecting the transcription factors *SOX2* and *OTX2* represent the most common causes of AMC, explaining 10–15% and 2–5% of cases, respectively[2], these conditions are genetically highly heterogeneous, with at least 130 genes currently included in standard structural eye disorders diagnostic panels (for example, https://panelapp.genomicsengland.co.uk/panels/509/). Despite

significant advances in our understanding of the genetic architecture of these disorders, approximately 40% of severely affected individuals remain genetically undiagnosed[2,3], likely to be due to analyses focussing on single nucleotide and copy number variants (SNVs and CNVs) in coding regions[2,4], thus missing pathogenic variants in introns and regulatory regions.

In humans, both monoallelic and biallelic *MAB21L2* (*Mab21-Like 2*) coding variants have been reported in individuals with AMC with and without skeletal anomalies, with arginine 51 an apparent hotspot impacted by 5/11 reported alterations[3,5–10], generally leading to a more severe phenotype of AMC and skeletal dysplasia[5,6,11]. Model systems have also highlighted the importance of *MAB21L2* in eye development. While mice with a heterozygous *Mab21l2* null mutation appear normal, homozygotes die by E14.5 with severe body wall defects and eye abnormalities, including cell proliferation defects in the optic vesicle, and absence of retinal pigment epithelium and lens[12]. In contrast, mice with the heterozygous missense change p.Arg51Cys exhibit anophthalmia/microphthalmia and shortened limbs similar to the phenotype observed in affected humans carrying this change[11]. Similarly, zebrafish *mab21l2* mutants, including knockdown and premature termination variants, display a variety of eye phenotypes, including microphthalmia, coloboma, small or absent lens, and misshapen optic cups[6,13–15].

Consistent with these findings, murine *Mab21l2* is strongly expressed in multiple tissues, including the eye, brain, heart, maxillary and mandibular process, limb bud and developing digits[12,16–18]. Zebrafish and *Xenopus* show similar expression in the developing eye, brain, limb buds and pharyngeal arches, but with some notable differences, including the specific expression pattern in the eye[13,19–21]. Similarly, *Mab21l2* expression is important for retinogenesis in the chick, with anophthalmia observed in knockdown animals[22].

Relatively little is known about the MAB21L2 function. There is evidence that Mab21l2 antagonises Bmp4 in *Xenopus*, and interacts directly with SMAD1, in both instances potentially acting as a transcriptional repressor or co-repressor[21]. In addition, multiple binding partners in humans and zebrafish, including TNPO2, KLC2, SPTBN1, HSPA5 and HSPA8, have been identified[23]. In contrast, a more detailed picture of the transcriptional regulation of the gene is emerging. *Mab21l2* appears to be a direct target of Pax6 (itself regulated by BMP4) in mouse lens, with two putative binding sites within its promoter[24]. Importantly, variants in both *BMP4* and *PAX6* have also been associated with AMC[25,26]. In zebrafish, the 7.7 kb sequence upstream of *mab21l2* regulates its expression in multiple tissues, including the forebrain, neural tube, branchial arches, lens and retina[27]. Furthermore, as *MAB21L2* and *MAB21L1* are a duplicated gene pair with overlapping patterns of expression[17,20], a comparison of their flanking regions in the mouse identified five evolutionarily conserved non-coding elements (Ma to Me)[28]. *LacZ* reporter analysis in mice demonstrated that four of these five elements direct tissue-specific expression during development, including in the midbrain (Ma), neural tube (Mb), branchial arches, otic vesicles and developing eye (Md), and spinal cord (Me)[28].

Here, we report two variants affecting *MAB21L2* in two families with AMC: a novel heterozygous missense variant (p.[Trp113Ser]), expanding the mutational spectrum of coding changes associated with AMC, and a 113,580 bp homozygous deletion ~19 kb upstream of the gene. By comparing the deleted region across species, we identify 15 non-coding conserved elements (CEs). Mouse ChIP-seq data show that two of these (CE13 and CE14) bind the transcription factor Otx2, an important protein for eye formation, while deletion of the region in both zebrafish and *Xenopus tropicalis* results in developmental eye anomalies. Moreover, targeted loss of the Otx2 binding site in CE14 in *Xenopus tropicalis* leads to smaller eyes and ocular coloboma, supporting the crucial role of this element in eye development. Our findings begin to unravel the upstream regulators for *MAB21L2* and reveal the importance of regulatory regions in human AMC, with significant impacts on clinical genetic diagnostics.

## Results
### Cohort screen
Sequencing of the coding region of *MAB21L2* in 835 individuals with AMC identified two individuals with likely pathogenic variants in this gene. The first case, previously described in Aubert-Mucca et al.[7], carried the heterozygous variant NM_006439.5:c.1 A > C (chr4:151504182 [hg19], p.[Met1?]). The second case (Family 1, Individual II.1) carried the heterozygous missense variant NM_006439.5:c.338 G > C (chr4:151504519 [hg19], p.[Trp113Ser]). This variant was absent from gnomAD (v2.1.1) and dbSNP153, predicted pathogenic by PolyPhen-2[29], SIFT[30] and METARNN[31], and had a CADD score of 32 (CADD model GRCh37-v1.6, https://cadd.gs.washington.edu/snv)[32]. The proband (Fig. 1A, Individual II.1) presented with bilateral microphthalmia and iris, choroid and retinal coloboma, right cataract, left congenital aphakia, and slightly delayed speech. His half-brother (II.2) was heterozygous for the same variant, presenting with bilateral iris, choroid and retinal coloboma, with nystagmus, bifid uvula, laryngomalacia, and mild neurodevelopmental delay (walking at 22 months). The variant was inherited from their mother (I.2), who had a small unilateral optic disc coloboma, microstomia and delayed speech. Comparative genomic hybridisation (array CGH) of the mother was normal. The variant was classified as likely pathogenic according to the VarSome annotation of the ACMG guidelines[33,34] (PM2 [supporting], PP1 [supporting], PP3 [strong]).

Genome-wide CNV screening of 188 UK individuals with AMC identified one individual (Family 2, III.5) with a homozygous deletion of between 107 kb and 122 kb on chromosome 4 located within *LRBA* and upstream of the nested gene *MAB21L2* (Fig. 1B). Long-range PCR and Sanger sequencing refined the CNV to chr4:151,370,119-151,483,698 (hg19) (113,580 bp) (Fig. 1C). This region is 19.38 kb upstream of *MAB21L2* and includes 4 exons of *LRBA*, resulting in a 150 amino acid in-frame deletion of all four *LRBA* isoforms, partially removing the BEACH-type PH and BEACH domains. The proband's mother (II.2) was heterozygous for the deletion; paternal DNA was unavailable for testing. SNP data for III.5 indicated that the CNV lies within a run of homozygosity (ROH) (Supplementary Fig. 1A), raising the possibility of segmental uniparental isodisomy since the parents were unrelated. SNP data analysis using PLINK showed that the ROH in III.5 extends for about 6 Mb and overlaps with a shorter ROH of approximately 1.3 Mb detected in the mother (II.2). Analysis of WES data for II.2 and III.5 confirmed this shared region of homozygosity, detecting a ROH of ~5 Mb in III.5 and of ~3 Mb in the mother. The difference in the estimated ROH size is due to the different number and density of variants analysed with the two methods. ROHs greater than 3 Mb in human populations are relatively rare[35]. The possibility of this ROH residing in an ROH hotspot was excluded by assessing 500 participants from the 1000 Genomes Project[36]. This detected a total of 6 ROH > 500 kb, and none matched the size of the ROH detected in III.5 (Supplementary Fig. 1B).

Individual III.5 (Family 2) is a female born by normal delivery at 39.5 weeks' gestation with a birth weight of 2.4 kg (2nd centile), length of 45.7 cm (0.4th-2nd centile) and head circumference of 30.5 cm (< 0.4th centile) to unrelated parents after a pregnancy complicated by concerns about intrauterine growth retardation and a postpartum haemorrhage. Bilateral microphthalmia was first noted at birth. At 4 weeks, there was no response to light, and an ophthalmological examination revealed very small orbits with microphthalmic remnants. Ultrasound confirmed bilateral microphthalmia: right globe 4.3 mm diameter and left globe 6.8 mm with evidence of retinal detachment and bilateral orbital cysts. She received sequential socket expansion using hydrophilic expanders, followed by solid prostheses. At 23 months she was noted to have corneal opacities and, additionally,

hearing impairment, generalised hypotonia and significant developmental delay (starting to sit unaided and speaking about four words at 23 months). She had dysmorphic features, including micrognathia, prominent transverse crura of her ears, upslanting palpebral fissures, an upturned nose, prominent philtral pillars, and frontal bossing. Her palate, spine, genitalia, limbs, hands and feet, as well as cardiovascular and abdominal examinations, were normal. At 12.5 years, she exhibited a significant learning disability, but no hearing impairment. She was

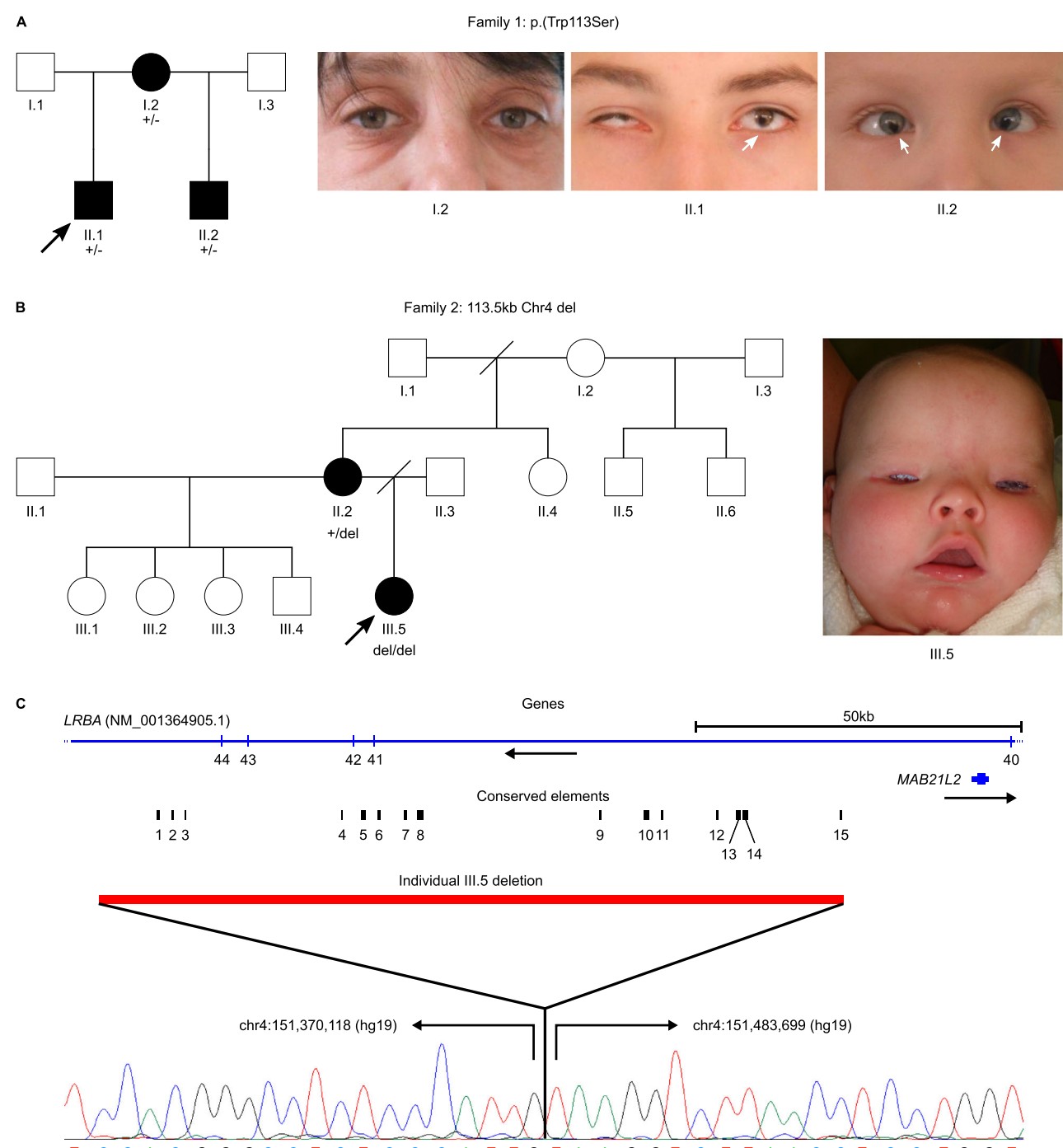

**Fig. 1 | *MAB21L2* variants in individuals with bilateral microphthalmia, including the genomic context of the deletion identified in Family 2. A** Pedigree and phenotypic images of the eyes of affected members of Family 1. The proband (II.1, black arrow) shows bilateral microphthalmia and iris coloboma (and has choroid and retinal coloboma, right cataract and left aphakia). His half-brother (II.2) shows bilateral iris coloboma (and has choroid and retinal colobomas, with nystagmus). Their mother (I.2) has a normal external eye appearance (with a small unilateral optic disc coloboma). All three individuals are heterozygous for the missense variant *MAB21L2* p.(Trp113Ser). **B** Pedigree of Family 2. The proband (III.5, black arrow) presented with bilateral microphthalmia (wearing early socket expanders in the photo), orbital cysts and microcephaly. A short upturned nose and prominent philtral pillars can be seen. Her mother (II.2) had very subtle bilateral central anterior lens suture opacities; her father was unavailable. III.5 is homozygous for an ~113.5 kb deletion ~19 kb upstream of *MAB21L2*, inherited from her heterozygous mother. **C** Schematic image of the deleted region identified in Family 2 showing relative positions of *LRBA* and *MAB21L2* (blue, with *LRBA* exon numbers indicated underneath), evolutionary non-coding conserved elements (CEs) within the deleted region, CNV location (red) and Sanger sequencing of proband demonstrating the breakpoints (sequencing of the breakpoint was performed for the negative strand only, for clarity the chromatogram shown is the reverse complement). CEs 7 and 8 contain the regulatory sequences Ma and Mb, respectively, identified by Tsang et al.[28].

**Table 1 | Non-exonic CEs identified within the region of the CNV identified in Family 2**

| Conserved Element | Position (hg19) | Length (bp) | Conservation to human orthologue (%) | | | |
|---|---|---|---|---|---|---|
| | | | Mouse | Chicken | *Xenopus* | Zebrafish |
| 1 | chr4:151,378,979-151,379,314 | 336 | 78.1 | 64.5 | – | – |
| 2 | chr4:151,381,225-151,381,449 | 225 | 86.7 | – | – | – |
| 3 | chr4:151,407,162-151,407,292 | 131 | – | – | – | – |
| 4 | chr4:151,410,152-151,410,335 | 184 | – | – | (42.9) | – |
| 5 | chr4:151,410,386-151,410,727 | 342 | – | – | (39.8) | – |
| 6 | chr4:151,412,674-151,413,028 | 355 | 73.2 | 73.0 | – | – |
| 7[a] | chr4:151,416,673-151,417,093 | 421 | 79.1 | 59.8 | 35.2 | 26.5 |
| 8[b] | chr4:151,418,673-151,419,605 | 933 | 85.6 | 67.3 | 58.2 | – |
| 9 | chr4:151,446,445-151,446,701 | 257 | 78.6 | – | – | – |
| 10[c] | chr4:151,453,193-151,453,960 | 768 | 88.2 | 78.9 | 59.9 | 31.9 |
| 11 | chr4:151,455,895-151,456,111 | 217 | 45.2 | 66.2 | 56.4 | – |
| 12 | chr4:151,464,211-151,464,504 | 294 | 91.2 | 54.2 | – | – |
| 13 | chr4:151,467,297-151,468,022 | 726 | 84.4 | 71.3 | 52.4 | – |
| 14 | chr4:151,468,356-151,469,080 | 725 | 82.4 | 72.3 | 43.4 | – |
| 15 | chr4:151,483,297-151,483,432 | 136 | 77.3 | – | – | – |

[a] Element 7 contains the regulatory sequence Ma[28]. [b] Element 8 contains the regulatory sequence Mb[28]. [c] Element 10 contains the regulatory sequence Mc[28]. Note: elements 4 and 5 do not map to the vicinity of *mab21l2* in *Xenopus tropicalis*.

treated for a chest infection at age 15 years, but prior to this had not shown an increased susceptibility to infection. Her mother (Individual II.2, Family 2) presented with subtle opacities of anterior crystalline lens suture, a history of total anomalous pulmonary venous drainage, and learning difficulties. The father was reported to have significant learning difficulties, but was unavailable to contact. Four maternal half-siblings had no ocular anomalies.

WES identified no other plausible pathogenic variants to explain the phenotype of Individual III.5 (Family 2) (Supplementary Information). While their homozygous deletion is predicted to cause an in-frame loss of 150 amino acids of *LRBA*, autosomal recessive *LRBA* deficiency is associated with immune function disorders, not developmental eye disorders[37,38]. In contrast, coding variants in *MAB21L2* are associated with ocular phenotypes, including AMC[3,5-10]. Given the presence of microphthalmia and coloboma in Individual III.5, we further investigated the impact of the deletion on *MAB21L2* function.

**Bioinformatic CNV analysis**
Two tissue-specific regulatory elements (Ma and Mb) identified by Tsang et al.[28] lie within the CNV region. As their analysis was restricted to elements conserved between murine *Mab21l1* and *Mab21l2*, we investigated the region to identify further *MAB21L2*-specific CEs.

Based on the "100 vertebrates Conserved Elements" track of the UCSC Genome Browser, we identified 15 non-coding CEs of at least 100 bp in length within the region deleted in Family 2. We mapped the genomic regions corresponding to the human CNV in mouse, zebrafish, *Xenopus tropicalis*, and chicken using comparative genome analysis (Supplementary Fig. 2 and Supplementary Table 1). This revealed that of the 15 non-coding CEs in the vicinity of *MAB21L2*, 12 were conserved in the mouse, 9 in chicken, 6 in *Xenopus tropicalis*, and 2 in zebrafish (Table 1). Two (CE7 and CE8) contained the tissue-specific regulatory elements Ma and Mb, respectively, while a third (CE10) spans a sequence conserved between mouse *Mab21l1* and *Mab21l2* (Mc)[28] (Table 1 and Supplementary Fig. 2).

**Modelling the CNV in zebrafish**
To assess the impact of the CNV, we initially modelled the deletion in zebrafish. We developed two permanent lines: *mab21l2^mw715^*, which carries a 32 kb deletion 12,140 bp upstream of *mab21l2* and corresponds to the CNV in Individual III.5 (chr1:24,032,155-24,064,147

[GRCz11]), and *lrba^mw716^*, carrying an indel variant in *lrba* exon 40 resulting in premature truncation (c.6454_6467delins; p.[Val2152*]) and disrupting *lrba* only. Both lines were viable with normal survival of all genotypes.

Progeny of *lrba^mw716^* heterozygous crosses developed no visible phenotypes. In contrast, at 24hpf 23.7% of embryos (41/173) from *mab21l2^mw715^* heterozygous crosses displayed variable small lens and misshapen eyecups with a visible gap between the neuroectodermal layers of the optic vesicle (ventral coloboma), of which 40/41 were homozygous for the deletion (Fig. 2A). At 48hpf homozygous *mab21l2^mw715^* fish also exhibited a visibly smaller optic tectum (Fig. 2A). All features (small lens, ventral coloboma, and small optic tectum) appeared transient and were not visibly detectable after 48hpf (small lens, ventral coloboma) or 72hpf (small optic tectum). Additional ocular measurements demonstrated a statistically significant reduction in lens diameter in *mab21l2^mw715^* homozygous embryos at both 24hpf ($p < 0.000001$) and 48hpf ($p = 0.001282$), but not at 72hpf ($p = 0.121573$) (Fig. 2B), with a subtle difference in eye width detected at 72hpf ($p = 0.028665$), but not at 24hpf ($p = 0.520299$) or 48hpf ($p = 0.300519$) (Fig. 2C).

Zebrafish homozygous for the *mab21l2^mw702^* knockout allele p.Gln48Serfs*5 display eye phenotypes including microphthalmia, coloboma, and small or absent lens[6]. Therefore, to determine whether *mab21l2^mw715^* represents a dysfunctional *mab21l2* allele, we bred compound heterozygote zebrafish carrying both the *mab21l2^mw715^* upstream deletion and *mab21l2^mw702^* coding loss-of-function (LOF) variant. Compound heterozygous *mab21l2^mw715/mw702^* fish displayed a similar transient small lens, ventral coloboma and small optic tectum phenotype to homozygous *mab21l2^mw702^* animals, supporting a role for the deleted region in *mab21l2* regulation (Fig. 2A).

**Spatiotemporal analysis of *mab21l2* gene expression in *mab21l2^mw715^* homozygous zebrafish embryos**
To explore the impact of the deletion on *mab21l2* transcriptional activity, we studied *mab21l2* expression in wild-type and mutant embryos using in situ hybridisation and quantitative RT-PCR (qRT-PCR); expression of *foxe3*, a gene involved in lens formation[39,40], was also tested. Consistent with previous studies, in situ hybridisation at 20-, 24- and 48hpf detected strong expression of *mab21l2* in the developing lens and retina in both WT and *mab21l2^mw715^* homozygous

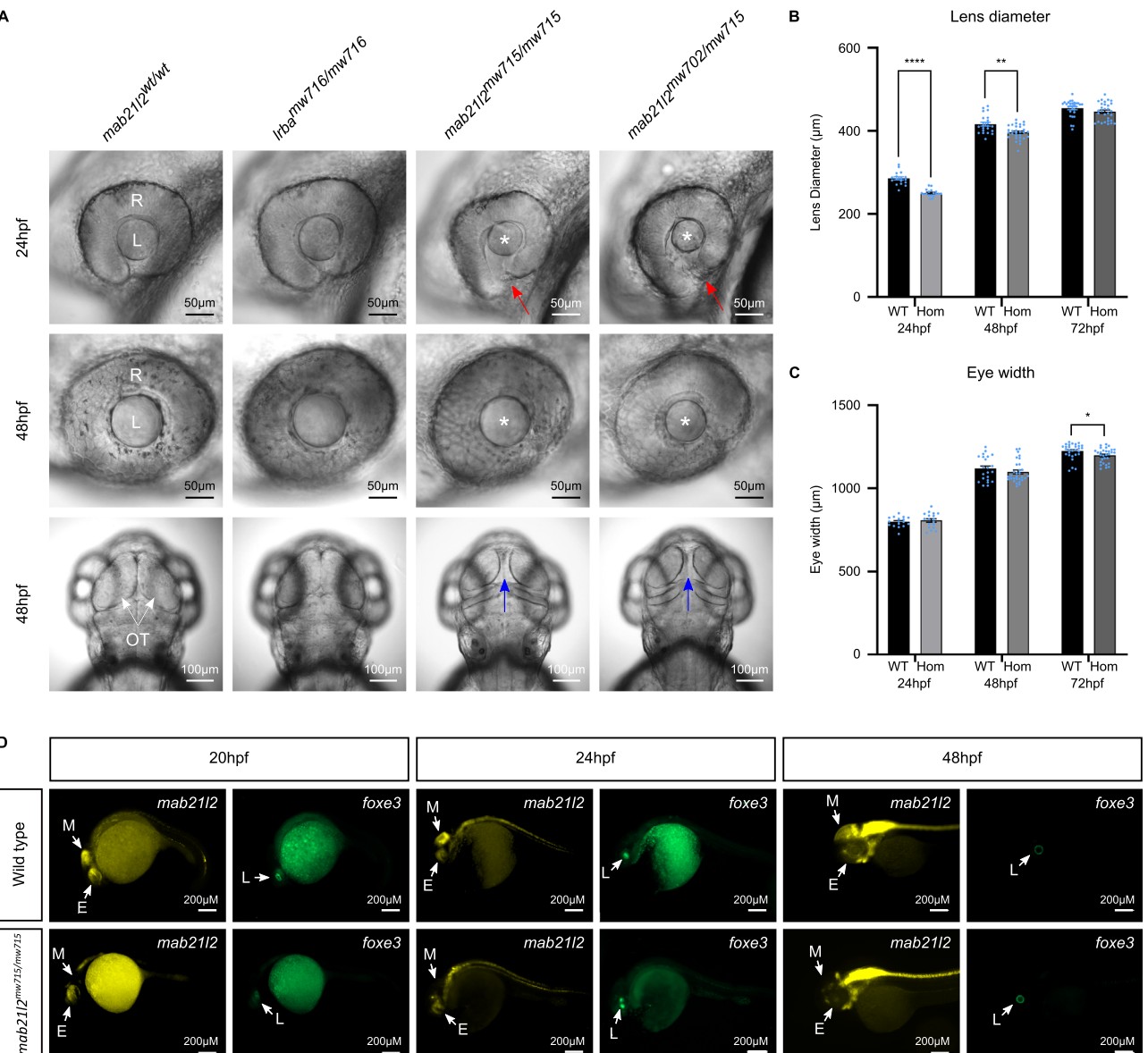

**Fig. 2 | Zebrafish modelling of the chromosome 4 CNV showing disruption in eye and midbrain development. A** Comparative images of the eye in wild-type (*mab21l2*^wt/wt^), *lrba* homozygous knockout fish (*lrba*^mw716/mw716^), fish with a homozygous deletion of the CNV identified in Individual III.5 (Family 2) (*mab21l2*^mw715/mw715^) and fish compound heterozygous for the deletion and a p.Gln48Serfs*5 frameshift allele in *mab21l2* (*mab21l2*^mw715/mw702^). Red arrows: misshapen eyecups with a visible gap between the neuroectodermal layers of the optic vesicle (ventral coloboma); white arrows: optic tectum; blue arrows: gap resulting from the reduced size of optic tectum; asterisks: small lens; R: retina; L: lens; OT: optic tectum. **B, C** Histogram of the lens diameter (**B**) and eye width (proximal to distal) (**C**) of wild-type and homozygous *mab21l2*^mw715^ fish at 24hpf (n = 17 and 18, correspondingly), 48hpf (n = 22 and 30) and 72hpf (n = 29 and 30). Lens diameter was significantly reduced in mutant fish at 24hpf (p < 0.000001) and 48hpf (p = 0.001282),

but not at 72hpf (p = 0.121573). Eye width was subtly decreased in mutant fish at 72hpf (p = 0.028665), but not at 24hpf (p = 0.520299) or 48hpf (p = 0.300519). Statistical analyses: two-tailed *t* test for two independent samples with Welch correction for unequal variances. **D** *mab21l2* (yellow) and *foxe3* (green) mRNA expression in whole mount wild-type and homozygous *mab21l2*^mw715^ fish. White arrows indicate the position of the midbrain (M), eye (E), and lens (L): the mutants exhibited a visibly reduced *mab21l2* expression in the midbrain region, and a slightly reduced *mab21l2* staining in the eyes at 20- and 24hpf; *foxe3* expression (green) in the developing lens was detected at all stages, showing a smaller lens in 20- and 24hpf mutant embryos. Panels (**B** and **C**): Black bars: wild-type; greyscale bars: homozygous *mab21l2*^mw715^ embryos (24hpf: medium grey; 48hpf: grey; 72hpf: dark grey); * p ≤ 0.05, ** p ≤ 0.01, ***p ≤ 0.001, ****p ≤ 0.0001. Source data is provided in the Source Data file.

embryos, with slightly diminished staining in 20- and 24hpf mutants (Fig. 2D). The subsequent qRT-PCR experiments using RNA from homozygous *mab21l2*^mw715^ ocular samples identified a reduction in *mab21l2* levels compared to WT at 24-, 48- and 72hpf, but this difference was not statistically significant (p = 0.1545, p = 0.4053, p = 0.0844, respectively; Supplementary Fig. 3A). In situ hybridisation of *mab21l2*^mw715^ homozygous embryos also demonstrated weaker *mab21l2* expression in the midbrain/optic tectum at 20hpf through

48hpf (Fig. 2D), consistent with the regulatory sequence Ma[28] being located within the deleted region. Subsequent qRT-PCR identified a significant reduction in *mab21l2* levels in 48hpf (p = 0.0002) and 72hpf (p = 0.003) homozygous midbrains compared to WT levels (Supplementary Fig. 3B). Overall, qRT-PCR using RNA from whole embryos suggested a decrease in *mab21l2* expression at early (20- and 24hpf), but not later (48- and 72hpf) stages. However, these differences were not statistically significant (p = 0.0551, p = 0.1177, p = 0.899, p = 0.7452,

respectively; Supplementary Fig. 3C). Expression of *foxe3* was detected by in situ hybridisation in both WT and small mutant lenses at all examined stages (20-, 24- and 48hpf) (Fig. 2D).

### Mouse ChIP-seq

Given the impact of the deletion on zebrafish development, we hypothesised that the non-coding CEs within the human CNV may contain transcription factor binding sites (TFBSs). Using quantitative ChIP-seq with an antibody to OTX2 in murine stem cells genetically modified to overexpress *Rax* (*Retina and Anterior Neural Fold Homeobox*, OMIM*601881), we identified three binding peaks within the region of the mouse genome corresponding to the human CNV. These were: chr3-2510 (chr3:86,571,497-86,571,891 [mm10]), chr3-5331 (chr3:86,572,696-86,572,877 [mm10]), and chr3-6127 (chr3:86,566,333-86,566,574 [mm10]) (Supplementary Fig. 4). Interestingly, chr3-5331 and chr3-2510 are located within CEs 13 and 14, respectively, present in humans, mouse and *Xenopus tropicalis*, but absent from zebrafish. In contrast, the sequence corresponding to chr3-6127 was not conserved in any of the other species under investigation (human, zebrafish, and *Xenopus tropicalis*), highlighting the potential for species-specific gene regulation.

### CE14 is a putative enhancer of *mab21l2* expression, mediated by Otx2 binding

Inspection of the regulatory landscape of the CNV region identified CEs 13 and 14 as putative retinal enhancers, each having an overlapping histone modification suggestive of enhancers (H3K27ac, H3K4me1) and mouse ChIP-seq OTX2 binding peaks (Supplementary Fig. 5A). Subsequently, we assessed possible TFBSs in both elements using TRANSFAC[41,42]. Consistent with our mouse ChIP-seq data, this identified a conserved OTX2 binding site within CE14. Analysis of CE14 using Xenbase[43] indicated that this region has the epigenetic marks of a poised enhancer at developmental Nieuwkoop and Faber (NF) stage 10.5, and that of an active enhancer during late gastrulation and mid neurulation (NF stages 12.5 and 16). This is consistent with the initiation of *mab21l2* expression during *Xenopus tropicalis* development[44] and the start of *Xenopus tropicalis* eye field specification at NF stage 12.5. Together, these data indicate that CE14 is a putative enhancer of *mab21l2* expression, mediated by Otx2 binding (Supplementary Fig. 5B).

### Targeting functional elements upstream of the *mab21l2* gene induces eye defects in *Xenopus tropicalis* embryos

Given the pivotal role of OTX2 in eye development, we used CRISPR-Cas9 to investigate the role of CE14 in *Xenopus tropicalis* by generating crispants with targeted disruption to the centre of the consensus Otx2 binding site (*mab21l2-CE14*). In parallel, we established crispants with a 38.5 kb deletion upstream of *mab21l2* corresponding to the deletion in Family 2 (*mab21l2-5′del*) (Supplementary Fig. 6). Eye defects were evident as early as NF stage 37 in both *mab21l2-CE14* and *mab21l2-5′del* animals (Fig. 3A). In both models, observed phenotypes included misshapen eyes and lens defects (Fig. 3A, B), being more severe in *mab21l2-5′del* animals. These included ocular coloboma as a result of incomplete optic fissure closure in approximately 85% of *mab21l2-5′del* crispants (33/39) and ~67% of *mab21l2-CE14* animals (35/52), at NF stage 41. In contrast, almost all animals targeted with a sgRNA designed to a region of the CNV locus harbouring no known TFBS (*mab21l2-non-CE*) had normal eye development (20/21) indistinguishable from WT animals (Fig. 3A). Furthermore, at NF stage 41 the size of the eyes in both *mab21l2-CE14* and *mab21l2-5′del* animals was significantly smaller than that of WT tadpoles; conversely, the eye size of *non-CE* crispants was comparable to that of WT clutchmates (one-way ANOVA followed by Tukey's multiple comparisons test: *non-CE* vs WT $p = 0.7274$; *CE14* vs WT $p = 0.0013$; *mab21l2-5′del* vs WT $p = 0.0007$, *non-CE* vs *CE14* $p = 0.0083$, *non-CE* vs *mab21l2-5′del* $p < 0.0001$, *CE14* vs *mab21l2-5′del* $p = 0.9814$) (Fig. 3C).

### Deep learning for 2D and 3D phenotyping of malformed eyes in CE14 crispants using light-sheet microscopy

To obtain a three-dimensional view of the ocular malformations, we fluorescently labelled NF stage 38 bilaterally microinjected *mab21l2-CE14* and *mab21l2-non-CE* crispants for Atp1a1 (renal tubules and neural tissue), the lectin peanut agglutinin (PNA; cone matrix domains) and wheat germ agglutinin (WGA; matrix domains surrounding rods). Light-sheet microscopy was then performed using the mesoSPIM platform (Fig. 3D)[45,46]. No gross developmental abnormalities were detected, with the exception of a decrease in volume and sphericity of *mab21l2-CE14* crispant retinas. Moreover, the combination of stains provided clear differentiation of organs (*i.e.* pronephros, neural tissue, developing eyes) (Fig. 3D, i and vii) in developing *Xenopus* embryos, serving as a robust dataset for training supervised deep learning networks, such as U-Net[47,48]. We extracted local three-dimensional context from each eye (Fig. 3D, ii and viii) and trained Mab-Net, our deep learning solution, for automated segmentation of the retina (Fig. 3D, vi and xii, cyan) and lens (Fig. 3D, vi and xii, yellow), facilitating straightforward feature extraction and automated size measurements. Mab-Net three-dimensional reconstruction of the retina and lens revealed a coloboma-like phenotype in *mab21l2-CE14* crispants, also confirmed by histological analyses of unilaterally injected animals showing optic fissure closure defects and malformed lens (Supplementary Fig. 7). In contrast, the *mab21l2-non-CE* group displayed an organised retina layering and a normal closure of the optic fissure (Fig. 3D and Supplementary Movie 1). Finally, surface reconstructions revealed a significant decrease in retinal volume and sphericity in the *mab21l2-CE14* group compared to the *non-CE* group (Fig. 3E, F). Together, these data further support a dependency on Otx2 binding in CE14 for normal ocular development and morphogenesis.

### CRISPR-mediated targeting of the CE14 element affects *mab21l2* mRNA expression in neurula-stage *Xenopus* tropicalis embryos

To investigate whether the abnormal eye morphogenesis observed upon disruption of the Otx2 binding site in the CE14 enhancer was associated with reduced *mab21l2* expression, we first examined the mRNA expression levels in whole embryos by qRT-PCR analysis. In WT animals, whole embryo *mab21l2* mRNA levels increased between NF stages 12.5 and 18, then remained relatively stable before another significant increase between NF stages 28 and 40 (Fig. 4A), consistent with published RNA-seq data[44]. Subsequent RT-PCR analysis on NF stage 20 CRISPR-targeted embryos showed that *mab21l2* transcript levels were significantly decreased in *mab21l2-CE14* crispants compared to both wild-type embryos and *non-CE* crispants (Fig. 4B). Since CE14 resides within an intron of *lrba*, its expression was also assessed. However, *lrba* mRNA expression levels were not affected by targeting of the CE14 element (Fig. 4B).

To determine whether the binding of Otx2 to the CE14 element affected specific subdomains of *mab21l2* expression, we first performed in situ hybridisation experiments in developing wild-type embryos (Fig. 4C). At late neurula stage (NF stage 18/19 and stage 20) expression of *mab21l2* is discernible in the optic field, the lens placode and a region in the midbrain. This expression pattern is also maintained at later stages (NF stages 22 and 23), with the appearance of expression in the hindbrain at early tailbud stage (NF stage 25). At mid tailbud stage (NF stage 28), *mab21l2* expression is predominantly expressed in the retina, midbrain, hindbrain and the neural crest derivatives (Fig. 4C). These expression patterns are similar to what has been previously reported for the related species *Xenopus laevis*[49]. Since Otx2 is already expressed in the anterior region of the early neurula, including the eye field, we investigated whether the earliest defined *mab21l2* expression domains (NF stage 18/19) were affected in unilaterally injected *mab21l2-CE14* crispant embryos. Upon the induced disruption of the Otx2 binding site in

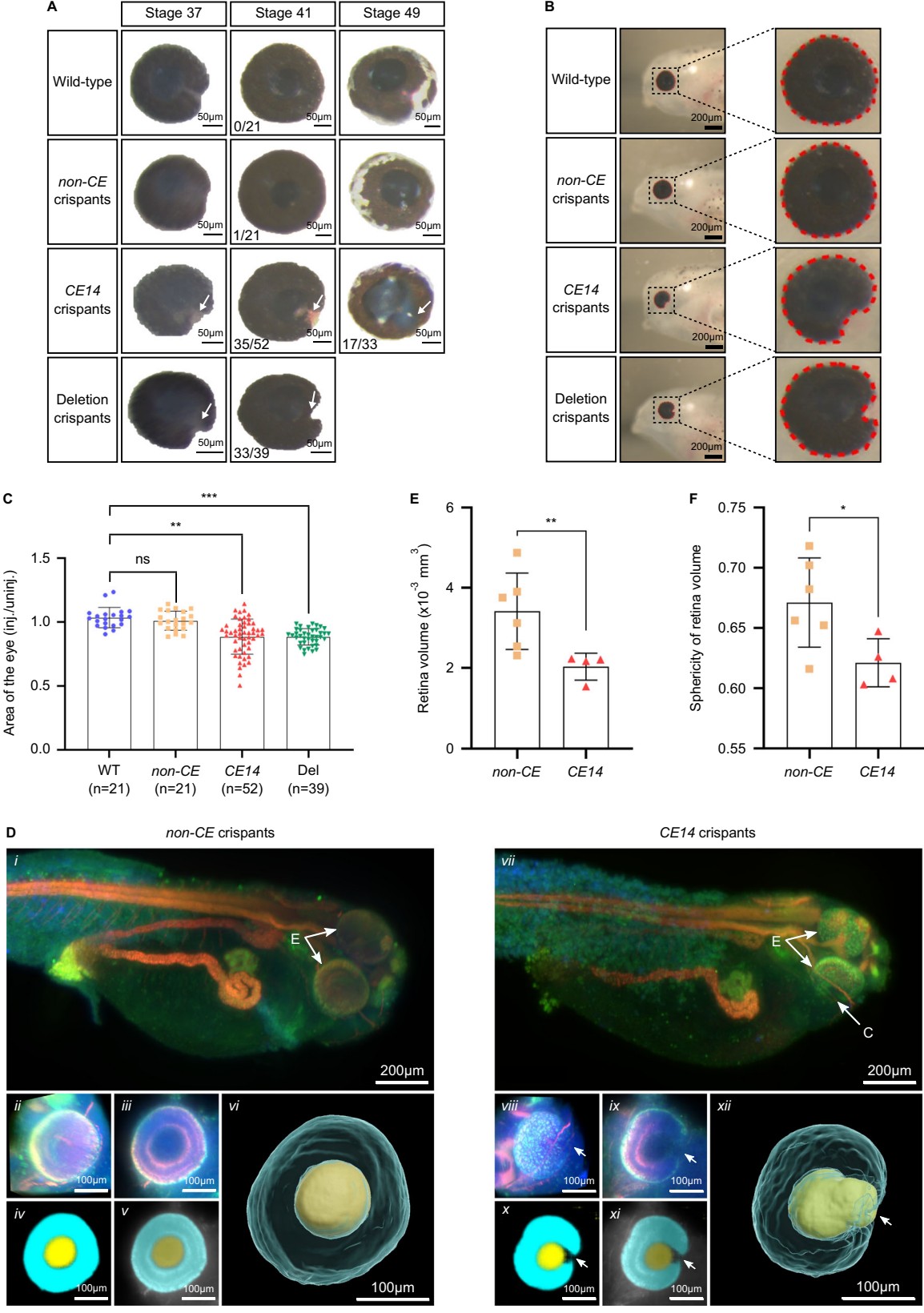

*CE14* crispants, embryos displayed a decreased *mab21l2* expression in the midbrain domain (14/20) and eye primordium (9/20). A diminished *mab21l2* expression in the lens placode was also observed (3/6). Conversely, *non-CE* crispants preserved normal expression in the eye primordium, lens placode and midbrain (9/10) (Fig. 4D).

## Discussion

Using data from humans, mice, zebrafish and *Xenopus tropicalis*, we demonstrate the importance of non-coding regulatory CEs in structural eye disorders and further elucidate the networks of genes orchestrating eye development. We describe two families with novel variants affecting the developmental gene *MAB21L2*. First is a missense

**Fig. 3 | Characterisation of eye size and morphology in *Xenopus tropicalis* *mab21l2* crispant embryos. A** Eye phenotypes in wild type (WT) and *mab21l2* crispant embryos throughout different developmental stages demonstrating colobomas (abnormal choroid fissure closure; white arrows) in *CE14* and deletion crispants. The number of animals showing the abnormal choroid fissure phenotype is indicated. **B** Outline of the circumference of the eyes (dashed red line) on the injected side of the tadpoles (NF stage 41) indicating the presence of coloboma in *CE14* and deletion crispants. **C** Histogram depicting relative eye size evaluated for individual embryos of wild-type (n = 21), *mab21l2-non-CE* (n = 21), *CE14* (n = 52) and *5'del* (n = 39) crispants via 2D measurements of the eye circumference. The ratio of the injected side versus the non-injected side of each group demonstrates a significant reduction in eye size on the injected side of *CE14* (p = 0.0013) and deletion crispants (p = 0.0007) compared to WT, while *non-CE* has a similar ratio (p = 0.7274). Statistical comparison of the eye size ratio, using data from three independent injections, was conducted using a randomised block experiment analysis in GraphPad Prism. Subsequently, mixed-effects analysis followed by

Tukey's post-test was employed to compare different experimental groups to the WT ratio. **D** 2D and 3D phenotyping of *mab21l2-CE14* and *mab21l2-non-CE* crispants. mesoSPIM light-sheet microscopy *in toto* imaging of a *mab21l2-non-CE* (i) and *CE14* (vii) embryo stained for PNA-Lectin (blue), WGA-Lectin (green) and Atp1a1 (red) showing coloboma in *CE14* crispants (abbreviations: C = coloboma, E = eye). Multiclass U-Net for segmentation of retina (cyan) and lens (yellow) are shown (ii-v, viii-xi), together with three-dimensional U-Net reconstructions (vi, xii) revealing aberrant retinal morphology, coloboma and malformed lens in *CE14* crispants (xii) (white arrows). **E, F** Histograms showing quantification of U-Net reconstructions of retina volume (**E**) and sphericity (**F**), revealing a significant reduction of both parameters in *mab21l2-CE14* crispant eyes (n = 4) compared to *non-CE* crispant eyes (n = 6). Statistical significance was determined using a non-parametric two-sided Mann-Whitney U test for the retina volume (p = 0.0095) and a two-sided unpaired *t* test for sphericity (p = 0.0409). Panels (**C**, **E** and **F**): Data are presented as mean ± SD; ns: not significant, *p ≤ 0.05, **p ≤ 0.01, ***p ≤ 0.001. Source data provided in the Source Data file.

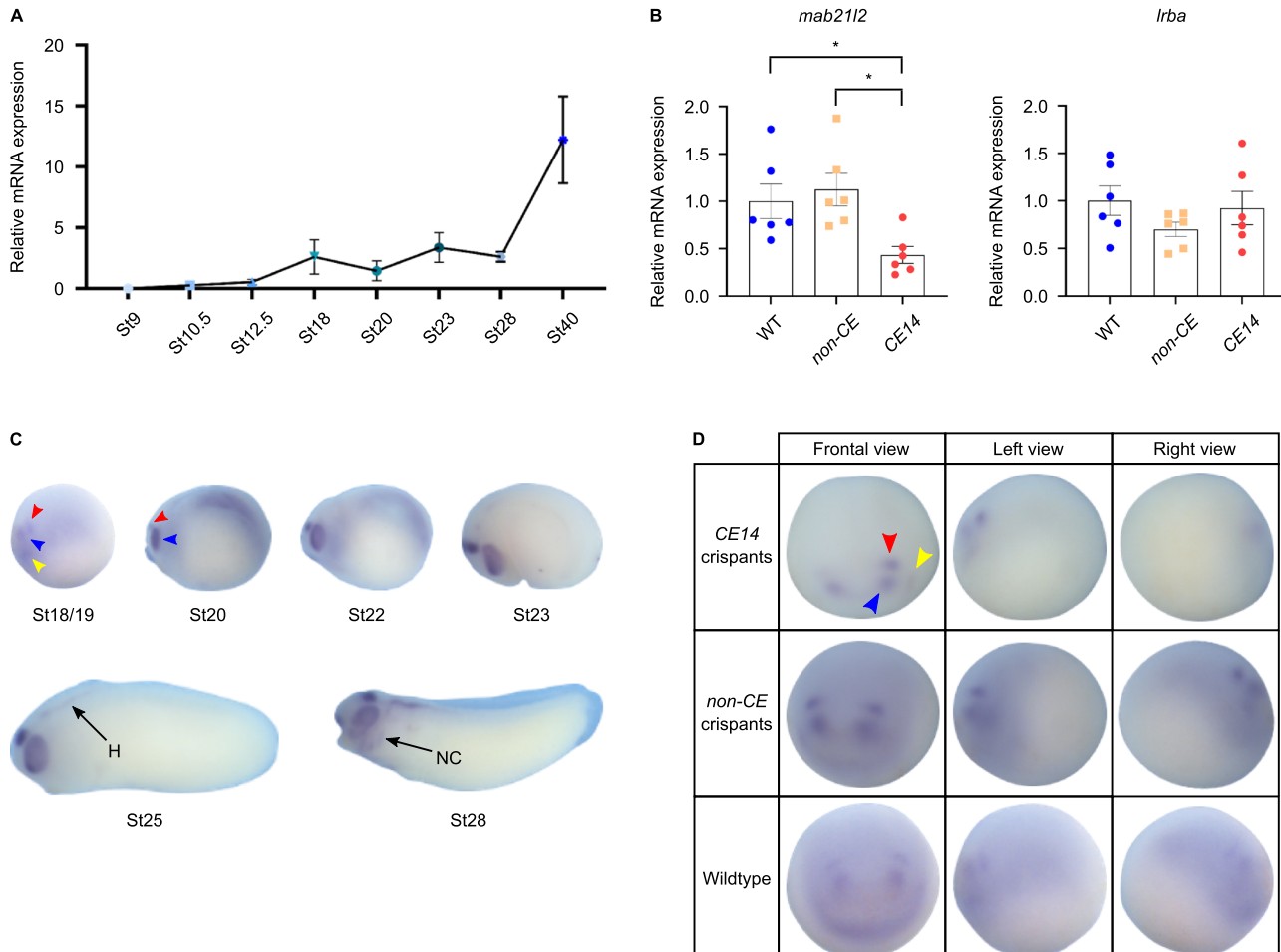

**Fig. 4 | Expression studies of *mab21l2* in wild-type, *CE14* and *non-CE* crispant *Xenopus tropicalis*. A** Quantitative RT-PCR analysis showing *mab21l2* expression levels in wild-type animals through different developmental stages: NF st9 (n = 6), st10.5 (n = 6), st12.5 (n = 6), st18 (n = 5), st20 (n = 5), st23 (n = 6), st28 (n = 6) and st40 (n = 5). Each data point represents mean ± SD. **B** Histograms demonstrating significantly reduced expression of *mab21l2* mRNA in *mab21l2-CE14* crispants compared with both *non-CE* crispants and wild-type embryos at NF stage 20 (*CE14* versus *non-CE*: p = 0.0166, n = 6; *CE14* versus wild-type: p = 0.0499, n = 6). In contrast, *lrba* expression was not affected (*CE14* versus *non-CE*: p = 0.5211, n = 6; *CE14* versus wild-type p = 0.3164, n = 6). For both genes, statistical significance was determined using one-way ANOVA followed by Tukey's post-test. Bars represent mean ± SEM; *p ≤ 0.05. **C** Whole mount in situ hybridisation throughout different developmental stages in wild-type animals showing *mab21l2* mRNA expression in

the optic field (blue arrowhead), the lens placode (yellow arrowhead) and a region in the midbrain (red arrowhead). Expression becomes apparent in the hindbrain (H) and the migrating neural crest cells (NC) later in development. Embryos are shown in a lateral view, anterior to the left. **D** In situ hybridisation showing expression of *mab21l2* mRNA in wild-type and crispant embryos at NF stage 18/19. Crispant embryos were injected at the 2-cell stage unilaterally on the right with *mab21l2-CE14* or *non-CE* gRNAs. Frontal, left and right views are depicted, with the left view representing the uninjected side. Patterns of *mab21l2* expression in the optic vesicle, lens placode or midbrain (blue, yellow and red arrowheads) are absent or reduced on the injected side for *mab21l2-CE14* crispants (upper panel) compared to *non-CE* crispants (middle panel) and wild-type (lower panel). Source data for panels 4A and 4B are provided in the Source Data file.

variant (c.338 G > C; p.[Trp113Ser]) segregating in a family with two half-siblings with microphthalmia, coloboma, lens anomalies, developmental delay and cleft uvula, and the mother with a milder ocular phenotype (optic disc coloboma), developmental delay and microstomia. Second, we report an individual with bilateral microphthalmia and a large homozygous deletion (~113.5 kb) approximately 19 kb upstream of *MAB21L2*, possibly the consequence of a maternal segmental uniparental isodisomy. The deleted region contains multiple evolutionarily conserved elements, including two known to drive tissue-specific expression in the mouse[28]. Modelling the deletion interrupted early eyecup, lens and optic tectum development, and significantly decreased *mab21l2* expression in the midbrain of zebrafish. Further, it led to microphthalmia, lens defects and coloboma in *Xenopus tropicalis*. Mouse ChIP-seq data revealed two previously unreported CEs within the locus which bind the AMC-associated transcription factor Otx2. Disruption of the Otx2 transcription factor binding site in one of these elements (CE14) in *Xenopus tropicalis* reduced expression of *mab21l2* in neurula stage embryos and resulted in microphthalmia, lens defects and coloboma, confirming its importance in eye development.

Of particular importance is our finding of the large homozygous deletion upstream of *MAB21L2*, which impacts gene regulation. Current knowledge of the role of regulatory enhancer elements in human developmental eye disorders is very limited. Individuals with variants in *PAX6*, a master control gene important for eye development, display severe eye anomalies, mainly aniridia, but also early onset cataracts, foveal hypoplasia, corneal changes and occasionally microphthalmia[50]. Variants affecting the SIMO region, an 800 bp key *PAX6* regulatory element approximately 150 kb downstream of the gene, have been reported in individuals with similar phenotypes, including aniridia[51,52], bilateral partial iris coloboma, congenital polar cataract and other related conditions[53], but not microphthalmia. Wormser et al. recently identified an *NHEJ1* intronic variant segregating with AMC in a large pedigree, with multiple affected individuals displaying phenotypes ranging from optic nerve coloboma to microphthalmia and anophthalmia[54]. Since the variant lies within a region capable of driving expression in the developing eye of chickens and mice, the authors suggested the region may be a putative enhancer for the neighbouring gene *IHH*, which is involved in animal eye development, but has not been linked to human structural eye disorders. Here, we report a deletion upstream of an established gene for human AMC, *MAB21L2*, and demonstrate that the loss of tissue-specific enhancer elements within the deleted region leads to microphthalmia. Therefore, this work contributes to an emerging body of evidence illustrating the significance of non-coding variants in human AMC. Moreover, the identification of an OTX2 binding site within the deletion provides new insights into the regulation of *MAB21L2* expression, helping delineate the gene interactions underlying AMC. The CNV does not involve *MAB21L2* or its immediate upstream promoter region[27]. However, multiple lines of evidence indicate that its impact on *MAB21L2* expression underlies the patient phenotype. In zebrafish, loss of the equivalent region resulted in transient small lens, misshapen eyecups, coloboma, and a smaller optic tectum, phenotypes comparable both to variants directly impacting the coding region of *mab21l2* [6,14,15] and knockdown of the gene[13]. The effect of the deletion on *mab21l2* was also confirmed by in situ hybridisation analyses, which showed a spatiotemporal decrease in its expression levels consistent with these transient defects. Similarly, our *Xenopus tropicalis* models of both the deletion and loss of the Otx2-binding site in CE14 had AMC phenotypes, including smaller eyes, lens anomalies and coloboma. Interestingly, retinal coloboma has been observed in several individuals with *MAB21L2* variants[5,6], including those in Family 1, with zebrafish[6,13] and mouse[12] models also presenting with aberrant retinal morphology. Our three-dimensional analyses of *mab21l2-CE14* crispant eyes showed a significant decrease in retinal volume and sphericity, providing a more

detailed picture of the ocular defects caused by the disruption of this element. Moreover, the disruption of CE14 led to a significant decrease of *mab21l2* (but not *lrba*) mRNA expression levels, further demonstrating the importance of OTX2 for *MAB21L2* regulation and normal eye development.

The use of multiple animal models in this study has been critical in deciphering both the mechanism by which the Family 2 CNV contributes to AMC and the complexity of the *MAB21L2* regulatory landscape. The zebrafish and *Xenopus tropicalis* models of the human deletion have different strengths and limitations. An important advantage of the zebrafish model is that it represents a stable genetic line where all cells in the developing mutant embryo carry the genomic deletion, similar to the affected Individual. Thus, it allows for a careful evaluation of the impact of the variant on development. However, several CEs identified in this study are not conserved in the zebrafish genome. In contrast, the *Xenopus tropicalis* genome demonstrates stronger conservation with the corresponding human region. While the second model is a mosaic crispant, it has permitted further dissection of the affected DNA sequences and identification of specific sites involved in the regulation of *MAB21L2/mab21l2* expression, in particular, the OTX2-binding CE14.

Interestingly, zebrafish exhibited only two of the conserved elements identified within the region of the deletion, CE7 and CE10. These encompassed the elements Ma and Mc, respectively, previously identified by comparing the *MAB21L1* and *MAB21L2* flanking sequences[28]. The loss of midbrain expression in our zebrafish model is consistent with the tissue-specific gene regulation observed for Ma in mice[28], demonstrating the evolutionarily conserved function of this element. In contrast, no known regulatory control function has been identified for Mc. Furthermore, CE14 is not present in zebrafish, which, as demonstrated by our mouse data, is able to bind Otx2, a transcription factor with a well-established role in eye development[2,55]. However, our zebrafish models also presented with transient AMC phenotypes. This suggests that either Mc has a species-specific role in zebrafish eye development or that in zebrafish there are additional *mab21l2* regulatory elements in the deleted region with functional relevance to eye development. The transient nature of the small lens phenotype observed in these models has at least two plausible explanations: other *mab21l2* regulatory regions might be able to restore lens expression of *mab21l2* following the initial deficiency; alternatively, its paralogue *mab21l1*, which is co-expressed with *mab21l2* in various tissues, could partly compensate for *mab21l2* dysregulation and ameliorate the phenotype. Our *Xenopus tropicalis* model also confirmed the complexity of the regulatory landscape of *mab21l2*. Disruption of CE14 results in structural eye anomalies, demonstrating the importance of this element for correct ocular development. However, the more penetrant phenotype in the *Xenopus tropicalis* deletion *versus* CE14 crispants suggests the involvement of additional regulatory elements within the region of the CNV. These may include CE13, which our mouse ChIP-seq data indicates also binds Otx2. Therefore, it remains important to further investigate these CEs to understand the tissue- and species-specific regulation of *MAB21L2* in development.

Although the CNV in Family 2 also results in an in-frame deletion of *LRBA*, this is unlikely to be relevant to their eye phenotype. Individuals with homozygous *LRBA* variants, including large deletions, can display immune system dysregulation, chronic diarrhoea, organomegaly, growth retardation and neurologic disease, typically beginning before 7 years of age, although onset has been reported as late as 17 years[37,38,56–58]. However, with the exception of one case with optic nerve atrophy[37,38,58], and another with 'bulging' of the left optic nerve[37,38,58], eye phenotypes have not been reported. Furthermore, developmental eye disorders are not reported in mouse *Lrba* knockouts[59], nor in the offspring of our zebrafish models that carried a premature truncation of *lrba*. As far as we are aware, our patient has none of the clinical features typically associated with LRBA deficiency. However, as

Individual III.5 is 15 years old, continued monitoring for phenotypes more typical of *LRBA* dysregulation is warranted.

While our study represents the first report of an upstream *MAB21L2* deletion resulting in microphthalmia, 9 SNVs in the coding region of the gene have been identified in conjunction with AMC and variable additional features[3,5–10]. Importantly, the *MAB21L2* missense variant identified in Family 1 contributes to delineating the emerging genotype-phenotype correlation observed for the previously reported SNVs. Changes affecting amino acids 49 and 51 typically lead to AMC and skeletal phenotypes, whereas those elsewhere in the gene are more likely to be associated with AMC that is isolated or in combination with milder anomalies. Consistent with this, the heterozygous missense variant identified in Family 1, affecting amino acid 113, is associated with milder extraocular features, not including skeletal anomalies. Among previously published variants, a single *MAB21L2* homozygous change (p.[Arg247Gln]) has been described in two brothers with AMC from a consanguineous family[5]. The elder brother presented with right retinal coloboma (including optic disc and macula), periorbital fullness, long eyelashes, epicanthus, prominent forehead, and a long and prominent philtrum. His younger brother had bilateral retinal coloboma (involving an optic disc on the right), right microphthalmia and exotropia, and facial dysmorphic features similar to his brother. Family 2 represents the second report of a homozygous variant affecting *MAB21L2* in an individual with developmental eye anomalies. Interestingly, the phenotypes described in Individual III.5 include some extraocular features overlapping those reported by the carriers of the homozygous missense change, such as prominent philtral pillars and frontal bossing. However, Individual III.5 also displayed phenotypes not previously observed in conjunction with *MAB21L2* variants, including micrognathia and microcephaly. Whilst it is possible that part of the phenotype may relate to *LRBA*, extraocular features may be explained by the effects of the deletion on specific regulatory elements upstream of *MAB21L2*.

*MAB21L2* is likely to have evolved as part of a duplication event with *MAB21L1*, adding a further degree of complexity to the genotype-phenotype correlations with regard to the CNV described in Family 2. Compensatory expression of *MAB21L1* may ameliorate the loss of *MAB21L2* expression in specific tissues in our patient, a hypothesis supported by the partially overlapping expression patterns of these genes in mice[17,20]. Indeed, reciprocal compensation has been suggested to explain the lack of overt brain phenotypes in both mouse *Mab21l1* knockouts and *Mab21l2* mutants[12,17]. During eye development, however, *MAB21L1* may not be able to fully compensate for *MAB21L2* dysregulation. In view of this, we note that III.5's heterozygous mother presents with bilateral central anterior lens suture opacities, in addition to learning difficulties. Given the impact of the deletion on zebrafish lens development and the heterozygous nature of pathogenic variants typically reported for *MAB21L2*, it is possible that the mother may be manifesting mild ocular features as a result of the deletion.

In conclusion, we demonstrate the importance of *MAB21L2* conserved regulatory elements as a cause of human AMC. By demonstrating the impact of the loss of a conserved OTX2 binding site, we are starting to dissect the regulatory networks involved in these conditions. Together, these data support an emerging paradigm shift whereby human eye development research and clinical testing should be expanded beyond the range of coding variants to include gene regulatory elements.

## Methods

### Ethics approvals and consent to participate
A UK cohort of 430 individuals with ocular developmental anomalies was recruited as part of a national 'Genetics of Eye and Brain Anomalies' study (Cambridge East Research Ethics Committee REC 04/Q0104/129); 545 French cases with various ocular developmental defects were screened for causative genetic variations after they gave their informed consent according to French Law. Informed consent was obtained according to the tenets of the Declaration of Helsinki. The authors affirm that research participants from Families 1 and 2 or their guardians provided written informed consent for the publication of the images included in Fig. 1.

The care and use of zebrafish (*Danio rerio*) were approved by the Institutional Animal Care and Use Committee at the Medical College of Wisconsin (protocol ID AUA00000352_AA_18).

*Xenopus tropicalis* experiments were performed according to the guidelines and regulations of Ghent University, Faculty of Sciences, Ghent, Belgium. Approval for the experiments was granted by the Ethical Committee for Animal Experimentation at Ghent University, Faculty of Sciences (approval number EC2020-025).

### Cohort sequencing
Individuals were sequenced either by whole exome sequencing (WES), as previously described[60,61], whole genome sequencing (WGS), customised NGS panel sequencing including 119 ocular developmental genes[7], or Sanger sequencing using standard protocols (primers available upon request). Screening for *MAB21L2* coding variants was performed in 290 genetically undiagnosed individuals with AMC from the UK (WGS = 47, WES = 94, structural eye disorders gene panel = 32, Sanger sequencing = 117) and 545 individuals with AMC from France (all analysed with a structural eye disorders gene panel).

### Copy number variant screening
In the UK cohort, 270 individuals (188 probands, 76 parents, and 6 other family members) were screened for copy number variants (CNVs) using the Illumina Infinium Global Screening Array (containing ~700 K SNPs). CNVs were called using PennCNV[62] and QuantiSNP[63]. For each algorithm, we retained calls including at least 3 consecutive SNPs, a length ≥1 kb, and a confidence score (PennCNV) or log Bayes Factor (QuantiSNP) of ≥10. A set of 'high confidence' CNVs was generated consisting of those detected by both algorithms with an overlap of at least 50%.

### Bioinformatic analyses
*Conservation:* The "100 vertebrates Conserved Elements" track of the UCSC Genome Browser was used to identify non-exonic regions with a minimum length of 100 bp, of which 75% of the sequence comprised CEs. The co-ordinates of these sequences in mouse, zebrafish, chicken and *Xenopus tropicalis* were identified using the Multiz alignment track. Conservation of the sequences was assessed using pairwise EMBOSS Needle alignment to the human sequence (https://www.ebi.ac.uk/jdispatcher/psa/emboss_needle), and Clustal Omega alignments of sequences available across multiple species (https://www.ebi.ac.uk/jdispatcher/msa/clustalo).

*Homozygosity mapping:* Runs of homozygosity (ROH) in individuals II.2 and III.5 (Family 2) were identified from SNP data using PLINK, searching for continuous stretches of at least 50 homozygous autosomal markers and a minimal length of 1 Mb, and from WES vcf data files using an in-house pipeline (https://github.com/sclokie/roh). Regions of ROH detected using WES data were compared to an unselected cohort of individuals from the 1000 Genomes Project. The VCF files included in the analysis are listed at https://github.com/sclokie/roh. Briefly, ROH of greater than 500 kbp from 500 individuals extracted from the 1000 Genomes Project were selected and visualised using a karyotype plot. We generated a summary plot to show the chromosome-wide ROH distribution, displayed alongside the ROH for Individuals III.5 and II.2 (Supplementary Fig. 1B).

### ChIP-seq
We performed quantitative ChIP-seq in murine stem cells genetically modified to overexpress *Rax* (*retina and anterior neural fold homeobox*) (CCE-Rx cells, gifted by Prof. S. Watanabe), which are able to differentiate into retinal ganglion cells[64]. A total of $2 \times 10^6$ resuspended

CCE-Rx cells were cultured in LIF-free medium on 10 cm bacterial plates. After 48 h, CCE-Rx embryoid bodies were treated with formaldehyde for 10 min, chromatin prepared, and ChIP performed according to the Upstate (Millipore) protocol, using 10 μg of anti-OTX2 antibody (abcam, ab21990-100), anti-SOX2 (Santa Cruz Biotechnology, sc-17320), or mouse IgG (Millipore, PP54) as a control. ChIP-seq libraries were prepared and sequenced using the standard Illumina protocol. The ChIP-seq data have been deposited to the GEO database (GEO accession number: GSE241711, https://www.ncbi.xyz/geo/query/acc.cgi?acc=GSE241711; NCBI BioProject: PRJNA1009210, https://www.ncbi.nlm.nih.gov/sra/PRJNA1009210). Peaks were called with the software Homer (http://homer.ucsd.edu/homer/, version available in 2022) using the following settings: region size of 180 bases, a minDist of 400 between peaks and a fdr of 0.00001.

### Zebrafish husbandry
Housing, care, breeding and staging of zebrafish embryos were performed using standard protocols[65].

### Generation and characterisation of zebrafish mutant lines
Zebrafish mutant lines were generated with the Alt-R CRISPR-Cas9 System (Integrated DNA Technologies, USA) using previously published protocols[66] and the following target-specific crRNAs: ATTTG-CATAAGGATACAAGA and TGTTGTCGACCACTTCCCAG for 5′ and 3′ positions of the deletion upstream of *mab21l2*, respectively, as well as TGGAGGTCTTCATGGCCAAC targeting exon 40 of the zebrafish *lrba* transcript.

The mosaic embryos were raised to adulthood, bred, and the resultant embryos screened by PCR amplification with primers flanking the targeted deleted region upstream of *mab21l2* (forward: CCCTAGCCATACGTAAACCTGT, reverse: TTTGTGCAATTTCAG-CAAGCCA) or the target site within exon 40 of *lrba* (forward: TTTGGCTTAAAGCAAGCTGAGG, reverse: TTAATCCTGCTCTGC-GACC). For both lines, the obtained PCR products were Sanger sequenced to verify the generated variants and obtain the exact coordinates of the *mab21l2* deleted region. Upon submission to ZFIN, the *mab21l2* upstream deletion allele was designated mw715 (ZFIN ID ZDB-ALT-230512-1) and *lrba* coding frameshift allele mw716 (ZFIN ID ZDB-ALT-230512-2). The *mab21l2*^mw702 zebrafish line (ZFIN ID ZDB-ALT-150611-1), carrying the frameshift variant Gln48Serfs*5, was previously described[6].

Morphological assessments and imaging of embryos were performed as previously described[66]. For lens diameter and overall eye size measurements, the embryos were photographed in the lateral (24- and 48hpf) or dorsal (72hpf) positions. Measurements were performed with ZEISS ZEN microscope software (Carl Zeiss) for 17 wild-type and 18 mutant eyes (at 24hpf), 22 wild-type and 30 mutant eyes (at 48hpf) and 29 wild-type and 30 mutant eyes (at 72hpf). A two-tailed t-test for two independent samples with Welch correction for unequal variances was used to determine whether there was a significant difference between the means of the lens diameter or eye width measurements in WT and homozygous groups at indicated time points. Standard error was calculated to determine the variance within each group. A more detailed description of the statistical analyses can be found in the Supplementary Information.

### Zebrafish expression studies
Whole-mount in situ hybridisation with RNAscope probes for *mab21l2* (499581-C2) and *foxe3* (499631) was performed using RNAscope Fluorescent Multiplex Detection Reagent Kit (Advanced Cell Diagnostics, cat#323100) according to the manufacturer's protocol.

For qRT-PCR expression assays, at least three independent RNA samples per developmental stage were extracted from whole embryos (10−12 pooled embryos per sample), dissected embryonic eyes (30−40 pooled eyes per sample) and midbrains (20−25 pooled brains per sample) using Direct-zol RNA MiniPrep kit (Zymo Research, cat#R2052). To remove DNA contamination, 1 μg of RNA was digested with 1 unit of DNaseI (ThermoFisher, cat# EN0523) for 15 min at room temperature (RT) and inactivated in the presence of 2.5 mM EDTA for 10 min at 65°C. Reverse transcription was performed using the SuperScript III First-Strand Synthesis System (ThermoFisher, cat# 18080051). Quantitative RT-PCR was performed using SYBR Green PCR Master Mix (Applied Biosystems, Waltham, MA, cat# 43-676-59) and CFX384 Touch Real-Time PCR Detection Systems (Bio-Rad, Hercules, CA). The primer sequences were: *mab21l2*, ACAATTCCAA-CACTCACGGC (forward) and AATCTCCGCGCATCATGTCA (reverse) (101 bp PCR product); *beta-actin*, GAGAAGATCTGGCATCACAC (forward) and ATCAGGTAGTCTGTCAGGTC (reverse) (323 bp PCR product).

All qRT-PCR experiments included a minus-reverse transcriptase control to exclude genomic DNA contamination. Reactions were performed in triplicate (technical replicates), and average Cq values were calculated. Each experiment included a minimum of three biological replicates. qPCR experiments with multiple peaks in a melt curve, indicative of non-specific amplification or primer-dimer formation, were excluded from further analysis. Total fold changes were estimated using the $2^{-\Delta\Delta Ct}$ method[67] with *beta-actin* expression as a normaliser. Technical replicates were averaged to generate a single value for each biological replicate. For both WT and homozygous samples, the mean and standard error of means (SEM) were calculated. A two-tailed unpaired *t* test was applied to the fold change values to determine statistical significance. Additional details regarding the statistical analyses can be found in the Supplementary Information. Graphs were generated using GraphPad Prism 9 (San Diego, California; https://www.graphpad.com/scientific-software/prism/).

### *Xenopus tropicalis* husbandry
Mating was induced in wild-type *Xenopus tropicalis* females and males by the injection of human chorionic gonadotropin (hCG) (Chorulon, MSD Animal Health) as previously described[68].

### Identification of putative cis-regulatory elements of *MAB21L2*
Comparative genome analysis was performed using the UCSC genome browser LiftOver tool between human and *Xenopus tropicalis* genomes. Putative *cis*-regulatory elements were determined based on the characteristics of enhancers, using the genome-wide multi-omics database RegNet (http://genome.ucsc.edu/s/stvdsomp/RegRet), ENCODE, ORegAnno, RefSeq functional elements and GeneHancer regulatory elements, datasets of histone modification marks (H3K27ac, H3K4me1) and ChIP-seq profiles of retinal transcription factors (CRX, NRL, OTX2), as previously described[69]. Furthermore, TFBS prediction analysis was performed for the conserved putative enhancers using TRANSFAC. The regions of interest were further assessed in a Xenbase (http://www.xenbase.org/, RRID:SCR_003280) merged dataset, using ChromHMM[70,71].

### Generation of mosaic mutant *Xenopus tropicalis* tadpoles using CRISPR/Cas9 genome editing
gRNAs were designed using the CRISPRScan software package[72]; the design strategy is illustrated in Supplementary Fig. 6A. A CRISPR gRNA (*CE14*) was designed to disrupt the Otx2 binding site in the putative enhancer site CE14: gaattaatacgactcactataggTCAATAATA-GAGGGATTAgttttagagctagaaatagc (Integrated DNA Technologies, USA). As a control, a gRNA (*non-CE*) directed against a sequence within the deletion region that lacks a potential TFBS was designed: gaattaatacgactcactataggCGCAAAGGATGGGTCGGGgttttagagctagaaatag (Integrated DNA Technologies, USA). Designed gRNAs were generated and quality controlled as previously described[73]. For the generation of the deletion upstream of *mab21l2* with paired AltR-crRNA (Integrated

DNA Technologies, USA), preparation of the gRNA complex was performed according to the manufacturer's protocol. Briefly, individual crRNAs targeting sites flanking the deletion region were annealed with AltR-tracrRNA separately, and 5 μM gRNA complex was prepared, combined with the in-house produced recombinant Cas9 protein and incubated for 10 min at 37 °C immediately prior to the microinjections. Following the individual incubations, the *CE14* and *non-CE* gRNA complexes were mixed for introduction into embryos. The coordinates of the gRNAs are provided in Supplementary Table 2.

### Phenotypic analysis of *mab21l2* mosaic mutant animals

For phenotypic analysis *Xenopus tropicalis* embryos were micro-injected unilaterally at the two-cell stage with 1 nl of a pre-incubated (1 min at 37 °C) mix of sgRNA (500–750 pg) and 1 ng of recombinant NLS-Cas9-NLS protein (VIB Protein Service Facility, UGent). To confirm efficient genome editing, pools of embryos at 3 days post fertilisation were lysed overnight in lysis buffer (50 mM Tris pH 8.8, 1 mM EDTA, 0.5% Tween-20, 2 mg/ml Proteinase K) at 55 °C and 800 rpm shaking, followed by 5 min incubation at 98 °C, and genotyped by PCR (Supplementary Table 3). The genome editing efficiency in F0 *Xenopus tropicalis* crispants was determined via targeted amplicon deep sequencing (Miseq, Illumina) and BATCH-GE analysis[74]. At NF stage 41, the area size of the eye of the injected and non-injected side of each animal was determined using ImageJ. Subsequently, the ratio between the eye size of injected and non-injected sides was calculated. Statistical comparison of eye ratio among distinct groups from three independent injections was conducted using a randomised block experiment analysis in GraphPad Prism. Subsequently, mixed-effects analysis followed by Tukey's post-test was employed to compare different experimental groups.

### Assessment of *mab21l2* mRNA expression level in different developmental stages

For the determination of *mab21l2* transcript levels, wild-type embryos at developmental stages including NF9, 10.5, 12.5, 18, 20, 23, 28 and 40, and different *mab21l2-CE14* and *non-CE* crispants (injected bilaterally at the 2-cell stage) were collected. Individual whole embryos were homogenised in the presence of TRIzol (catalogue number 15596026, Invitrogen), and total RNA was obtained by phenol-chloroform extraction. cDNA was generated using the iScript cDNA synthesis kit (Bio-Rad). Real-time amplification was performed using the Sensi-FAST™ SYBR® No-ROX Kit (Bioline) and a LightCycler® 480 Instrument II (Roche Life Science) with primer pairs spanning exon-exon junctions: for *mab21l2*, primer sequences were CACAGGTTTACCCCACCTCC (forward) and CCACTGTAATGGGGCCTTGT (reverse); for *lrba* GGCATTTTCTGTGTTTTTAGCA (forward) and TCTGATACCTGCCACACCAT (reverse). The housekeeping genes *slc35b*, *odc1*, *ipcat3* and *sub1* were used for normalisation. Relative gene expression levels were determined using the software qbase + (Biogazelle), with Normalised Relative Quantities (NRQs) calculated based on the housekeeping gene expression levels.

### In situ hybridisation of *mab21l2*

The developmental expression pattern of *mab21l2* was determined in wild-type *Xenopus tropicalis* and mosaic mutant embryos using in situ hybridisation. To generate the probe, two primers for *mab21l2*, each coupled to an RNA polymerase promoter sequence (T3 or T7), were designed in order to amplify an ~1000 bp region from a *Xenopus tropicalis* cDNA preparation (Supplementary Table 4). Following PCR cloning of the amplified fragment (NEB® PCR Cloning Kit), anti-sense and sense digoxigenin-labelled probes were transcribed using T3 and T7 RNA polymerases, respectively. Whole-mount in situ hybridisation was carried out under high stringency conditions at 60 °C using a previously described protocol[75].

### Immunostaining of *Xenopus tropicalis mab21l2* crispants

Whole-mount immunofluorescence was performed as previously described[76]. NF stage 38 tadpoles were fixed in 1 × MEMFA (10 ml 10× MEMFA salts, 10 ml 37% formaldehyde and 80 ml $H_2O$) for 30 min at RT, followed by 3 × 5 min PBS washes. Tadpoles were dehydrated and permeabilized with 4 × 10 min 100% methanol washes and kept in 100% methanol overnight at − 20 °C. Tadpoles were then bleached in 10% $H_2O_2$/23% $H_2O$/66% methanol overnight under strong light, then rehydrated with a series of 5 min washes with methanol/PBST (1 × PBS, 0.2% Tween 20) as follows: 75% methanol/25% PBST, 50% methanol/50% PBST, 25% methanol/75% PBST and 5 × 100% PBST. Tadpoles were blocked for 2 h in 10% Natural Goat Serum (NGS) in 0.2% PBST at RT and incubated with mouse anti-Atp1a1 (1:200, DSHB, A5)[77] and a lectin mixture diluted in 10% NGS overnight at 4 °C (Wheat Germ Agglutinin, Alexa Fluor™ 594 [WGA] [1:200 ThermoFisher, W11262], and Lectin PNA from Arachis hypogaea [peanut], Alexa Fluor™ 488 [1:200, ThermoFisher, L21409]). For nuclear counterstaining, DAPI (1:500, ThermoFisher, D1306) was added to the primary antibody/lectin mixture.

Tadpoles were washed with PBST for 30 min and blocked for 2 h at RT with 10% NGS in 0.2% PBST. The blocking buffer was replaced with goat anti-mouse IgG (H + L) secondary antibody (Dylight633) in 10% NGS and incubated for 2 h at RT. Tadpoles were then washed for 1 h with PBST and 1 h in PBS. For *Xenopus tropicalis* mesoSPIM imaging, embryos were embedded in 2% low-melting agarose and dehydrated as follows: 25% methanol/75% 1 × PBS (45 min), 50% methanol/50% 1 × PBS (45 min), 75% methanol/25% 1 × PBS (45 min), three times in 100% methanol (2 × 45 min, 1 × overnight). The clearing was performed in BABB (benzyl alcohol:benzyl benzoate 1:2) overnight.

### Microscopy and deep learning reconstructions

*In toto, Xenopus tropicalis* embryos were imaged using selective plane illumination microscopy (mesoSPIM)[45,46]. For all mesoSPIM recordings, fluorophores were excited with the appropriate laser lines and a quadband emission filter (ZET405/488/561/640, AHF) was employed. Embryos were imaged at a voxel size of $1 \times 1 \times 1\ \mu m^3$ (X × Y × Z) using an MVPLAPO1X objective (Olympus). From mesoSPIM recordings, sub-volumes (300x300xdepth) containing eyes were manually extracted. Network training was on a classical 2D U-Net architecture using the model architecture of 2D-CellNet and the U-Net Fiji plug-in[47,48]. Intersection Over Union (IOU) reported are as calculated by the U-Net Fiji plug-in, and a tile size of 508 × 508 pixels was used. Fluorescence recorded in the 488, 561 and 640 were used to train/deploy as a three-channel hyperstack. To avoid aberrant normalisation, we applied a small blob (1 × 1 pixels) in each z-slice that contained the median pixel intensity across the entire 3D recording and across all three channels of the hyperstack prior to training and deployment. Multiclass U-Net (retina and lens) was trained using 88 training images, obtained by taking each 30th section from each eye sub-volumes, and 21 validation images, obtained by taking 2 to 3 sections (slices spaced 50 sections apart in Z) from each eye sub-volumes. Both retina and lens were manually annotated by an expert as separate classes for training. We trained for 20,000 iterations at a learning rate of 1E-4, followed by 5000 at 5E-5 and 200 at 2E-5. We reached an IOU of 0.90 for the retina and an IOU of 0.73 for the lens. The softmax output layer was thresholded for binary mask conversion at 0.98 for the retina and at 0.5 for the lens. Binary masks were processed in Imaris (Bitplane), and volume/sphericity was extracted from three-dimensional surface reconstructions. Sphericity is calculated by the ratio of the surface area of the sphere to the surface area of the particle.

The U-Net Fiji plugin[47] (https://lmb.informatik.uni-freiburg.de/resources/opensource/unet/) can be found at the following github project page: https://github.com/lmb-freiburg/Unet-Segmentation. Pre-trained weight and model files for the U-Net model used in this study can be found at https://lienkamplab.org/deep-learning-models/.

## Reporting summary

Further information on research design is available in the Nature Portfolio Reporting Summary linked to this article.

## Data availability

Participant raw sequencing and genotyping data supporting this study are not publicly available due to their containing information which could compromise research participant privacy and consent. Informed consent in this work does not cover the deposition of full sequencing data from the participants to a repository. Access to specific subsets of processed data generated in this study can be obtained by request from Nicola Ragge (nragge@brookes.ac.uk) with a period for response to the request of one calendar month. Data can only be shared for research purposes with permission of the patient and/or legal guardian(s) and via data-sharing agreements. The processed data cannot be shared with third parties; if the data is to be used for scientific presentations and/or publications, the applicant should contact Nicola Ragge for agreement, and with agreed acknowledgement. Sanger sequencing data confirming the two variants described in this study have been deposited to SRA. The VCF files containing sequencing data generated by the 1000 Genomes Project and used in our ROH analyses are listed at https://github.com/sclokie/roh. The data supporting the zebrafish and *Xenopus tropicalis* findings are available within the paper and Supplementary Information files. The ChIP-seq data generated in this study have been deposited to the GEO database (GEO accession number GSE241711, https://www.ncbi.nlm.nih.gov/geo/query/acc.cgi?acc=GSE241711; NCBI BioProject: PRJNA1009210, https://www.ncbi.nlm.nih.gov/sra/PRJNA1009210). Pre-trained weight and model files for the U-Net model can be found at https://lienkamplab.org/deep-learning-models. Source data are provided in this paper.

## Code availability

Code generated for ROH analyses is available at https://github.com/sclokie/roh. Bash scripts for annotation and filtering of Family 2 sequencing data are provided as separate files (Supplementary Software 1 folder).

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

## Acknowledgements

We would like to thank the families for their generous participation in the study and clinicians who have kindly referred patients to the study, in particular Professor Richard Collin, Dr Alison Salt, and Mr Abou-Rayyah, Moorfields Eye Hospital, London. We are also grateful to Dr Ke Yan and Dr Jian Zhang for their consultation and help with statistical analyses related to zebrafish RT-qPCR data. This work was generously supported by grants from Baillie Gifford (F.C., R.H., L.V.T., D.A.B., Y.K. and N.K.R.), Microphthalmia, Anophthalmia and Coloboma Support (MACS) (www.macs.org.uk) (D.A.B.), HEIF (Health Innovation Fund, Oxford Brookes University) (N.K.R.), Thames Valley and South Midlands National Institute for Health and Care Research (NIHR) Clinical Research Network (CRN) (D.A.B.). French patients were recruited through the Rares Diseases Cohorts (RaDiCo) programme, which is funded by the French National Research Agency under the specific programme "Investments for the Future", Cohort grant agreement ANR-10-COHO-0003 (N.C., J.P. and P.C.). Funding was provided by National Institutes of Health grants R01EY025718 and R01EY034398 to E.V.S. This work was also supported by the Ghent University Special Research Fund (BOF20/GOA/023) (E.D.B. and K.V.); H2020 MSCA ITN grant (No. 813490 StarT) (E.D.B. and K.V.). E.D.B. is a Senior Clinical Investigator (1802220 N) of the Research Foundation-Flanders (FWO); M.B.C. was an Early Starting Researcher of StarT (grant No. 813490). E.D.B. is a member of ERN-EYE (Framework Partnership Agreement No. 739534-ERN-EYE). Funding was further provided by the European Union's Horizon 2020 research and innovation programme to T.N. (Marie Skłodowska-Curie grant agreement No. 891127) and S.S.L. (grant agreement No. 804474, DiRECT), and the Swiss National Science Foundation to S.S.L. (Project 310030_189102). Imaging was performed with equipment maintained by the Centre for Microscopy and Image Analysis (ZMB), University of Zurich. The authors also thank Dr José María Mateos Melero for mesoSPIM training and support.

## Author contributions

**Fabiola Ceroni:** Conceptualisation, Data curation, Formal analysis, Investigation, Methodology, Visualisation, Writing – review & editing. **Munevver B. Cicekdal:** Conceptualisation, Data curation, Formal analysis, Investigation, Methodology, Validation, Visualisation, Writing – review & editing. **Richard Holt:** Conceptualisation, Data curation, Formal analysis, Investigation, Methodology, Validation, Visualisation, Writing – original draft, review & editing. **Elena Sorokina:** Conceptualisation, Data curation, Formal analysis, Investigation, Methodology, Validation, Visualisation, Writing – review & editing. **Nicolas Chassaing:** Data curation, Funding acquisition, Investigation, Methodology, Project administration, Resources, Writing – review & editing. **Samuel Clokie:** Data curation, Formal analysis, Software, Writing – review & editing. **Thomas Naert:** Data curation, Formal analysis, Investigation, Software, Visualisation, Writing – review & editing. **Lidiya V. Talbot:** Data curation, Formal analysis, Software, Writing – review & editing. **Sanaa Muheisen:** Data curation, Investigation, Validation. **Dorine A. Bax:** Project administration, Writing – review & editing. **Yesim Kesim:** Data curation, Formal analysis. **Emma C. Kivuva:** Resources. **Catherine Vincent-Delorme:** Resources. **Soeren S. Lienkamp:** Funding acquisition, Project administration, Resources, Supervision. **Julie Plaisancié:** Resources. **Elfride De Baere:** Conceptualisation, Funding acquisition, Project administration, Resources, Supervision. **Patrick Calvas:** Funding acquisition, Project administration, Resources, Supervision. **Kris Vleminckx:** Conceptualisation, Funding acquisition, Methodology, Project administration, Resources, Supervision, Writing – review & editing. **Elena Semina:** Conceptualisation, Funding acquisition, Project administration, Resources, Supervision, Visualisation, Writing – review & editing. **Nicola K Ragge:** Conceptualisation, Funding acquisition, Project administration, Resources, Supervision, Visualisation, Writing – review & editing.

## Competing interests

The authors declare no competing interests.

## Additional information

¹Faculty of Health and Life Sciences, Oxford Brookes University, Oxford, UK. ²Department of Pharmacy and Biotechnology, University of Bologna, Bologna, Italy. ³Department of Biomedical Molecular Biology, Ghent University, Ghent, Belgium. ⁴Center for Medical Genetics, Ghent University Hospital, Ghent, Belgium. ⁵Department of Biomolecular Medicine, Ghent University, Ghent, Belgium. ⁶Department of Ophthalmology, Medical College of Wisconsin, Milwaukee, USA. ⁷Centre de Référence des Affections Rares en Génétique Ophtalmologique CARGO, Site Constitutif, CHU Toulouse, Toulouse, France. ⁸Service de Génétique Médicale, Hôpital Purpan, CHU de Toulouse, Toulouse, France. ⁹Institute of Anatomy, University of Zurich, Zurich, Switzerland. ¹⁰Zurich

Kidney Center, University of Zurich, Zurich, Switzerland. [11]Centre for Human Genetics, University of Oxford, Old Road Campus, Oxford, UK. [12]NIHR Oxford Biomedical Research Centre, John Radcliffe Hospital, Oxford University Hospitals NHS Foundation Trust, Oxford, UK. [13]Royal Devon University Healthcare NHS Foundation Trust, Exeter, UK. [14]Service de Génétique Clinique Guy Fontaine, Hôpital Jeanne de Flandre, Lille, France. [15]Centre de Biologie Intégrative (CBI), Centre de Biologie du Développement (CBD), Université de Toulouse, CNRS, UPS, Toulouse, France. [16]West Midlands Regional Clinical Genetics Service, Birmingham Women's and Children's NHS Foundation Trust and Birmingham Health Partners, Birmingham, UK. [17]These authors contributed equally: Fabiola Ceroni, Munevver B. Cicekdal, Richard Holt, Elena Sorokina. ✉e-mail: kris.vleminckx@irc.ugent.be; esemina@mcw.edu; nragge@brookes.ac.uk

