## [Peer Review File · Nature Communications]

Deletion upstream of MAB21L2 highlights the importance of evolutionarily conserved non-coding sequences for eye developmentReviewer #1 (Remarks to the Author):

The main noteworthy result in this report relates to the novel homozygous deletion upstream of MAB21L2. The examination of this in zebrafish and xenopus models reveals impact of deletion of regulatory elements of MAB21L2 in eye development. Detailed functional analysis of non-coding structural anomalies is helpful in gradually elucidating additional likely genetic causes of developmental eye diseases. This is a detailed and thorough report examining functional impact of this structural non-coding variant.

Reviewer #2 (Remarks to the Author):

In this manuscript, Ceroni et al. describe a novel deletion in a cis-regulatory region of the gene MAB21L2 which elicits similar phenotypes to coding mutations in the same gene, most notably microphthalmia and coloboma. The authors identify 15 elements within this region which are variably conserved across several vertebrates. Support for the human patient and bioinformatic data comes from experiments in a number of model systems. Disruption of homologous genomic regions in zebrafish and *Xenopus* produces similar phenotypes, albeit transiently in the former. The authors focus on a conserved element within the deleted region which they identify as a binding site for the transcription factor OTX2. Targeted disruption of the homologous sequence in *Xenopus* causes microphthalmia and coloboma, as well as disrupting *mab21l2* expression in optic primordia of embryos. This suggests regulation by OTX2 via a conserved cis-regulatory element is critical for MAB21L2 function in early eye development of humans and some other vertebrates.

The study presents a relatively rare example of a disease-causing regulatory element variant and its recapitulation in animal models. It also adds to our understanding of the regulation, and evolution thereof, of a poorly understood developmental gene. The experiments and analyses are mostly well executed, although there are a few issues in this regard, and the text is well written. It should be of general interest to the clinical genetics and developmental biology fields.

Major comments

Highly unequal sample sizes, such as those seen in Fig. 3C, severely compromise the robustness of a One-way ANOVA. A reliable conclusion as to the differences between groups cannot be drawn as a result. An increase in the sample sizes of the WT, non-CE and Del groups, so that they match the size of the CE14 group, is necessary. Very unequal sample sizes are also apparent in Fig. 3A and the 24 and 72 hpf groups in Fig. 2B and C.

The sample size difference in Fig. 3 also makes it difficult to determine if the more specific CE14 crispant has a milder phenotype than the full deletion crispant. Currently, it appears that the difference in eye size is quite small, which I find surprising.

The PCR results demonstrating efficient genome editing in *Xenopus* crispants (lines 629-632) should be included as a supplementary figure at least. It is important to include this data when using a crispant approach, as phenotypic variability can easily be due to incomplete mutagenesis. Ideally, the authors would provide some form of sequencing data with majority mutant reads of the targeted locus in a bilaterally injected crispant DNA sample. See for examples Kroll et al. Fig. 2 (doi:10.7554/eLife.59683), or Tornini et al. Fig 1-supplement 2 (doi:10.7554/eLife.82249). A column for "Efficiency" is present in supplementary table 2, but it is unclear how these numbers were derived and the last gRNA does not even have one.

Minor comments

In line 308-309 the authors state that *mab21l2* CE14 crispants have a "disorganised retinal structure, whereas the *mab21l2*-non-CE group displayed an organized retina layering". In what way is the retinal layering of CE14 crispants disrupted? Based on the staining in the images provided (Fig. 3D), retinal lamination appears normal in these larvae, with the exception that the optic fissure is open. Higher magnification/resolution images should be provided to support this claim and the exact nature of the disruption described. There are several examples of *mab21l2* zebrafish mutants in which retinal layering is at least partially intact.

Environmental stress often exacerbates developmental phenotypes in zebrafish. Did the authors try raising the *mab21l2mw715* mutants at 32°C or exposing them to some other environmental

perturbation? It might elicit a stronger effect.

In Fig.2D, the 48 hpf mutants appear to have elevated expression of mab2112 and foxe3 along the region of the optic fissure. Is this real expression or autofluorescence? Higher magnification images of the eyes/tectum would be better for this part of the figure, as it is difficult to compare the controls and mutants with the current images.

The authors show that the sphericity of mab2112 crispant retinas is decreased. Is this a known phenotype in humans or other species with mab2112 mutations?

Line 701: IOU is misspelled as IUO. IOU should be defined.

Reviewer #3 (Remarks to the Author):

The study reported shows that a deletion in the upstream region of the mab2112 gene found in a family with eye defects causes ocular coloboma when re-created in *Xenopus* and also does so (albeit transiently) in zebrafish. By combining mouse cell ChIP and frog crispants it was shown that a specific DNA element containing an *otx2* binding site was needed for proper expression of mab2112. This is an important series of experiments that demonstrates the power of using multiple models to identify links between non-coding gene variants and disease; something that is still relatively rare and demanding.

Overall the study is clearly presented, the data are strong and the experiments well controlled.

Science Comments:

- 1) In zebrafish, the decrease observed in mab2112 expression (fig 2E) is much greater in the brain than eye. Were there any effects seen or tested consistent with neurodevelopmental abnormalities? If not, might it be useful to note this at the point in the discussion made around compensation in the mouse (line 460)?
- 2) Why are the changes seen in zebrafish transient? Is this commonly seen, is it a reflection of high regenerative capacity? Please comment.
- 3) Overall the study takes advantage of the large numbers of individuals that can be used for studies like this in frog and fish with minimal ethical costs. In fig. 4D however, the numbers are low (3/6 and 2/6 showing loss of expression in the eye region, lines 342 and 343). Since this is an important link between CE14, mab2112 and potentially one element of the cause of disease it would be good to see the number of embryos tested increased.

Minor comments:

Since the *Xenopus* work took place in the EU should there be a licence number associated with it?

line 61 needs:20% of childhood....

Line 266:each having overlapping....

Line 293: should littermates not be replaced with clutchmates considering the animal is a frog?

Response to the reviewers

Here, we respond to the comments of our reviewers. Each comment is listed in turn, with our responses summarised in bullet points beneath each. The locations of all changes to the manuscript are indicated including page and line numbers, according to the tracked version.

Reviewer #1 (Remarks to the Author):

The main noteworthy result in this report relates to the novel homozygous deletion upstream of MAB21L2. The examination of this in zebrafish and xenopus models reveals impact of deletion of regulatory elements of MAB21L2 in eye development. Detailed functional analysis of non-coding structural anomalies is helpful in gradually elucidating additional likely genetic causes of developmental eye diseases. This is a detailed and thorough report examining functional impact of this structural non-coding variant.

Response

- We would like to thank the reviewer for their acknowledgement of both the value and rigour of our study. As there are no specific comments to be addressed, we have not made any additional changes to the manuscript in response.

Reviewer #2 (Remarks to the Author):

In this manuscript, Ceroni *et al.* describe a novel deletion in a cis-regulatory region of the gene *MAB21L2* which elicits similar phenotypes to coding mutations in the same gene, most notably microphthalmia and coloboma. The authors identify 15 elements within this region which are variably conserved across several vertebrates. Support for the human patient and bioinformatic data comes from experiments in a number of model systems. Disruption of homologous genomic regions in zebrafish and *Xenopus* produces similar phenotypes, albeit transiently in the former. The authors focus on a conserved element within the deleted region which they identify as a binding site for the transcription factor OTX2. Targeted disruption of the homologous sequence in *Xenopus* causes microphthalmia and coloboma, as well as disrupting *mab21l2* expression in optic primordia of embryos. This suggests regulation by

OTX2 via a conserved cis-regulatory element is critical for MAB21L2 function in early eye development of humans and some other vertebrates.

The study presents a relatively rare example of a disease-causing regulatory element variant and its recapitulation in animal models. It also adds to our understanding of the regulation, and evolution thereof, of a poorly understood developmental gene. The experiments and analyses are mostly well executed, although there are a few issues in this regard, and the text is well written. It should be of general interest to the clinical genetics and developmental biology fields.

Major comments

Highly unequal sample sizes, such as those seen in Fig. 3C, severely compromise the robustness of a One-way ANOVA. A reliable conclusion as to the differences between groups cannot be drawn as a result. An increase in the sample sizes of the WT, non-CE and Del groups, so that they match the size of the CE14 group, is necessary. Very unequal sample sizes are also apparent in Fig. 3A and the 24 and 72 hpf groups in Fig. 2B and C.

The sample size difference in Fig. 3 also makes it difficult to determine if the more specific CE14 crispant has a milder phenotype than the full deletion crispant. Currently, it appears that the difference in eye size is quite small, which I find surprising.

Response

- As suggested, we have enhanced the robustness of these experiments by incorporating two additional independent microinjections into the experimental setup. The compiled experiments again demonstrate that *CE14* and *del* crispants exhibit significantly reduced eye size ($p=0.0013$ and $p=0.0007$, respectively) compared to wild-type tadpoles. Conversely, the *non-CE* crispants demonstrate comparable eye size to the wild type ($p = 0.7274$). We utilized a randomized block experiment analysis in GraphPad Prism to address experiment-to-experiment variability. Mixed-effects analysis revealed significant effects of different crispant groups, while appropriately adjusting for variability between experiments, ensuring the robustness of our findings. Figures 3A and 3C have been modified accordingly, along with the accompanying legends (pages 42-43 lines 1082-1096), and information in the main text (page 14 lines 302-313).

- In addition, we have increased the number of samples used for measurements of the lens diameter and eye width of zebrafish (Figures 2B and 2C). These data provide increased support for the difference between wildtype and mutant fish lens diameter at 24hpf, and indicate a subtle difference in eye width at 72hpf. We have updated Figure 2 to illustrate this additional data, and altered the manuscript text in light of these findings (pages 10-11 lines 229-231).

The PCR results demonstrating efficient genome editing in *Xenopus* crispants (lines 629-632) should be included as a supplementary figure at least. It is important to include this data when using a crispant approach, as phenotypic variability can easily be due to incomplete mutagenesis. Ideally, the authors would provide some form of sequencing data with majority mutant reads of the targeted locus in a bilaterally injected crispant DNA sample. See for examples Kroll et al. Fig. 2 (doi:10.7554/eLife.59683), or Tornini et al. Fig 1-supplement 2 (doi:10.7554/eLife.82249). A column for “Efficiency” is present in supplementary table 2, but it is unclear how these numbers were derived and the last gRNA does not even have one.

Response

- In response to the reviewers concerns we have generated a new supplemental figure (Supplemental Figure 5). This figure includes an example of the relative contribution of different indels for CE14 and non-CE guide RNAs (Supplemental Figure 5C). Briefly, the genome editing efficiency in F0 *Xenopus tropicalis* crispants was confirmed via targeted amplicon sequencing and BATCH-GE analysis (Steyaert 2018 doi: 10.1007/978-1-4939-8784-9_6). DNA sequencing of each target region in non-CE and CE14 crispants, from a sample pool of three individual embryos lysed at NF stage 41, revealed editing efficiencies of 80% and 45%, respectively. Additionally, a comprehensive illustration demonstrating the design of guide RNAs for both the deletion and disruption of the Otx2 binding motif within the *CE14* putative enhancer is provided (Supplemental Figure 5A). In order to confirm the presence of the large 39 kb deletion in the injected embryos, we designed primers flanking the paired guide RNA target sites. We have incorporated a gel image exhibiting the presence of the deletion in the *deletion* crispants (Supplemental Figure 5B). In this gel image, DNA samples extracted from each experimental group were subjected to PCR amplification of the targeted region

using the designed flanking primers. Notably, the presence of the deletion enabled the amplification of a distinct 500bp PCR fragment. In contrast, the CE14, non-CE, and wild-type groups exhibited no amplification, corroborating the specificity and efficacy of our targeted deletion strategy.

- Supplementary Table 2 has been updated to include the editing efficiency of the guide RNAs altR1 gRNA (60%) and altR2 gRNA (22%).

Minor comments

In line 308-309 the authors state that mab21l2 CE14 crispants have a “disorganised retinal structure, whereas the mab21l2-non-CE group displayed an organized retina layering”. In what way is the retinal layering of CE14 crispants disrupted? Based on the staining in the images provided (Fig. 3D), retinal lamination appears normal in these larvae, with the exception that the optic fissure is open. Higher magnification/resolution images should be provided to support this claim and the exact nature of the disruption described. There are several examples of mab21l2 zebrafish mutants in which retinal layering is at least partially intact.

Response

- We thank the reviewer for this comment which prompted a deeper examination of retinal lamination in *CE14* crispants. Our initial interpretation was primarily based on the images obtained with the light sheet microscope, which did not provide the required cellular resolution to support our statement. As such, we have conducted additional histological analyses on *CE14* crispants. These experiments revealed preservation of normal retinal organization on the injected side of the tadpoles, with the evident anomalies such as an open optic fissure and malformed lens. Therefore, based on these new findings, we retract our earlier statement regarding retinal disorganization observed in previous experiments, and have amended the manuscript accordingly (Results: page 15, lines 330-334; Discussion page 19, lines 430). We also provide a new Supplemental Figure 6 with illustrative histological sections from an injected and non-injected eye from a *CE14* crispant.

Environmental stress often exacerbates developmental phenotypes in zebrafish. Did the authors try raising the *mab21l2*^{mw715} mutants at 32°C or exposing them to some other environmental perturbation? It might elicit a stronger effect.

Response

- We did indeed try raising *mab21l2*^{mw715} fish at higher temperatures, but observed no effect on the phenotype.

In Fig.2D, the 48 hpf mutants appear to have elevated expression of *mab21l2* and *foxe3* along the region of the optic fissure. Is this real expression or autofluorescence? Higher magnification images of the eyes/tectum would be better for this part of the figure, as it is difficult to compare the controls and mutants with the current images.

Response

- The reviewer is correct to point out the autofluorescence. As such, we have repeated these *in situ* hybridisation experiments and provided new images at higher magnification (Figure 2D) and with reduced autofluorescence to allow easier comparison of control and mutant eyes.

The authors show that the sphericity of *mab21l2* mutant retinas is decreased. Is this a known phenotype in humans or other species with *mab21l2* mutations?

Response

- We acknowledge the reviewer's comment of the novelty of our findings regarding the effects on sphericity associated with *mab21l2* mutations. To the best of our knowledge, such effects have not been previously reported in the literature. The sphericity of the retina holds significant importance as it directly contributes to the structural integrity and function of the visual system. Our decision to report on the aberrant sphericity observed upon disruption of the *Otx2* binding site stems from our 3D analysis on the lightsheet microscope. It is not known to us whether the absence of any similar reports on sphericity either means that such analysis has either not been performed in *MAB21L2* affected humans and other species, or whether such

phenotype is absent (or temporary) in these cases. However, previous studies have indicated that *Mab21L2* variants can impact retinal morphology more generally in animal models (zebrafish and mouse), in addition to causing retinal coloboma in humans. We have added a sentence with relevant references to the discussion to highlight this point (page 19, lines 426-428).

Line 701: IOU is misspelled as IUO. IOU should be defined.

Response

- We have corrected the spelling of IOU (page 33 line 763) and provided the definition (page 32 line 752).

Reviewer #3 (Remarks to the Author):

The study reported shows that a deletion in the upstream region of the *mab21l2* gene found in a family with eye defects causes ocular coloboma when re-created in *Xenopus* and also does so (albeit transiently) in zebrafish. By combining mouse cell ChIP and frog crispants it was shown that a specific DNA element containing an *otx2* binding site was needed for proper expression of *mab21l2*. This is an important series of experiments that demonstrates the power of using multiple models to identify links between non-coding gene variants and disease; something that is still relatively rare and demanding.

Overall the study is clearly presented, the data are strong and the experiments well controlled.

Science Comments:

1) In zebrafish, the decrease observed in *mab21l2* expression (fig 2E) is much greater in the brain than eye. Were there any effects seen or tested consistent with neurodevelopmental abnormalities? If not, might it be useful to note this at the point in the discussion made around compensation in the mouse (line 460)?

Response

- We have performed a preliminary 'embryonic touch response assay' (Kokel et al 2010; <https://pubmed.ncbi.nlm.nih.gov/20081854/>), and in addition did not find any gross

deficiencies at embryonic stages. It is possible that some specific deficiencies could be identified using additional tests and/or analysis of adult fish, however this is beyond the scope of this paper.

2) Why are the changes seen in zebrafish transient? Is this commonly seen, is it a reflection of high regenerative capacity? Please comment.

Response

- It is possible that other *mab21l2* regulatory regions could be activated in zebrafish, thus restoring lens expression of *mab21l2* following the initial deficiency and rescuing the early small lens phenotype. Another possibility is that there could be compensation by another member of the family, *mab21l1*, which is co-expressed with *mab21l2* in various tissues. There are other examples of transient phenotypes in zebrafish, including a report of transient hyperglycemia in *Dio2*KO zebrafish (Houbrechts et al., 2019). We have included a comment about this within the discussion (page 20 lines 449-453).

3) Overall the study takes advantage of the large numbers of individuals that can be used for studies like this in frog and fish with minimal ethical costs. In fig. 4D however, the numbers are low (3/6 and 2/6 showing loss of expression in the eye region, lines 342 and 343). Since this is an important link between *CE14*, *mab21l2* and potentially one element of the cause of disease it would be good to see the number of embryos tested increased.

Response

- We have conducted additional experiments to increase the sample size for the *in situ* experiments involving *CE14* and non-*CE* crispants. These experiments, comprising 13 additional embryos for *CE14* crispants and 7 for non-*CE* crispants, in essence confirmed our initial findings. Specifically, among the *CE14* crispants, 10 out of 13 embryos exhibited altered *mab21l2* expression, while non-*CE* crispants displayed normal expression patterns. The signal in the lens placode was not always discernable, most likely due to its very temporary expression or high background staining in some embryos. Hence, for the lens placode only a lower number of samples could be scored.

Consequently, we have updated Figure 4D and revised the description in the Results section accordingly (page 16 lines 367-372).

Minor comments:

Since the *Xenopus* work took place in the EU should there be a licence number associated with it?

Response

- All experiments involving *Xenopus tropicalis* were conducted following the guidelines and regulations established by Ghent University, Faculty of Sciences, Ghent, Belgium. Approval for the experiments was granted by the Ethical Committee for Animal Experimentation at Ghent University, Faculty of Sciences (approval number EC2020-025). The relevant information has been added to the manuscript (page 28 lines 647-649).

line 61 needs:20% of childhood....

Response

- As suggested, we have added "of" to the sentence (page 4 line 65).

Line 266:each having overlapping....

Response

- We have altered the relevant sentence to convey our meaning more clearly (page 13, line 280).

Line 293: should littermates not be replaced with clutchmates considering the animal is a frog?

Response

- As suggested, we have replaced littermates with clutchmates (page 14 line 309).

Additional alterations to the manuscript

During the process of our revisions in response to the reviewers comments we have made some further minor alterations to the manuscript to improve clarity, listed below. All changes to the text have been marked in the submitted manuscript:

- Introduction: we have updated the number of genes currently included in the UK R36.1 Structural Eye Disorders diagnostic panel (page 4 line 68).
- Figure 1
 - o We have permission to include an image for the proband of Family 2
- Figure 3
 - o The order of panels has been re-arranged to improve the clarity of the figure
 - o Individual images in Fig 3D have been labelled with Roman numerals to support interpretation via the main text and legend
 - o Two panels (*Dv* and *Dix*) have been replaced to provide more accurate information
 - o Scale bars included for each image in Figure 3D
 - o Additional labelling has been added to images in Fig 3D to aid interpretation
 - o Additional descriptions have been added to the legend and main text
- Supplemental Figure 2: The original figure mistakenly referred to the “Interval deleted in individual II.1”. This has now been corrected to read “Interval deleted in individual III.5).
- Figures (general): Minor changes to alignments of panels, label font sizes etc
- We have expanded the Discussion (page 18, lines 402-414) to acknowledge the publication of a recent paper (Wormser et al., 2023) describing the identification of an intronic variant disrupting a potential *IHH* regulatory element in a large AMC pedigree.
- We have also made some minor text changes to improve clarity.

Reviewer #2 (Remarks to the Author):

I am satisfied with the authors' responses and associated revisions. I believe they considerably improve the quality of the manuscript. I have no further comments or concerns.

Reviewer #3 (Remarks to the Author):

The resubmitted manuscript, which was already strong in its original form, has improved markedly. Particularly in the increased number of samples in key experiments and improved analysis. The minor comments have also been addressed.